Manuscript prepared for Atmos. Chem. Phys.
with version 2014/05/30 6.91 Copernicus papers of the LaTeX class copernicus.cls.
Date: 18 May 2016

# Derivation of physical and optical properties of midlatitude cirrus ice crystals for a size-resolved cloud microphysics model

**Ann M. Fridlind**[1]**, Rachel Atlas**[2]**, Bastiaan van Diedenhoven**[1,3]**, Junshik Um**[4]**, Greg M. McFarquhar**[4]**, Andrew S. Ackerman**[1]**, Elisabeth J. Moyer**[2]**, and R. Paul Lawson**[5]

[1]NASA Goddard Institute for Space Studies, 2880 Broadway, New York, NY, USA
[2]University of Chicago, Chicago, IL, USA
[3]Columbia University, New York, NY, USA
[4]University of Illinois, Urbana-Champaign, IL, USA
[5]Spec Inc., Boulder, Colorado, USA

*Correspondence to:* Ann Fridlind (ann.fridlind@nasa.gov)

**Abstract.**

Single-crystal images collected in mid-latitude cirrus are analyzed to provide internally consistent ice physical and optical properties for a size-resolved cloud microphysics model, including single-particle mass, projected area, fall speed, capacitance, single-scattering albedo, and asymmetry parameter. Using measurements gathered during two flights through a widespread synoptic cirrus shield, bullet rosettes are found to be the dominant identifiable habit among ice crystals with maximum dimension ($D_{max}$) greater than 100 $\mu$m. Properties are therefore first derived for bullet rosettes based on measurements of arm lengths and widths, then for aggregates of bullet rosettes and for unclassified (irregular) crystals. Derived bullet rosette masses are substantially greater than reported in existing literature, whereas measured projected areas are similar or lesser, resulting in factors of 1.5–2 greater fall speeds, and, in the limit of large $D_{max}$, near-infrared single-scattering albedo and asymmetry parameter ($g$) greater by $\sim$0.2 and 0.05, respectively. A model that includes commonly imaged side plane growth on bullet rosettes exhibits relatively little difference in microphysical and optical properties aside from $\sim 0.05$ increase in mid-visible $g$ primarily attributable to plate aspect ratio. In parcel simulations, ice size distribution and $g$ are sensitive to assumed ice properties.

# 1 Introduction

It is well known that cirrus clouds substantially impact radiative fluxes and climate in a manner that depends upon their microphysical and macrophysical properties (e.g., Stephens et al., 1990). With respect to microphysical properties, observations of cirrus cloud particle size distributions and underlying ice crystal morphology still remain subject to large uncertainties, in part owing to lack of instrumentation adequate to provide artifact-free and well-calibrated measurements of size-distributed ice particle number and mass concentrations (e.g., Baumgardner et al., 2011; Lawson, 2011; Cotton et al., 2012). With respect to single-crystal properties, the Cloud Particle Imager (CPI) instrument provides high-resolution images of crystals at 2.3 $\mu$m per pixel (Lawson et al., 2001), but to our knowledge no airborne instrumentation to date provides a direct measurement of the most fundamental quantity: single-particle mass. How important is advancement of such microphysics observations? On one hand, for instance, simulated climate sensitivity has been reported sensitive to cirrus ice fall speeds (e.g., Sanderson et al., 2008). On the other hand, statistical properties of cirrus simulated at the cloud-scale have been reported relatively insensitive to ice crystal habit assumptions (e.g., Sölch and Kärcher, 2011). Such an insensitivity to ice habit presents a contrast to mixed-phase cloud simulations, which are found sensitive to even relatively minor changes in the specification of ice microphysical properties such as habit, fall speed, and size distribution shape (Avramov and Harrington, 2010; Avramov et al., 2011; Fridlind et al., 2012; Ovchinnikov et al., 2014; Simmel et al., 2015).

It is also well known that ice crystals in the atmosphere exhibit a profound degree of diversity in morphology that impacts microphysical process rates and radiative properties (e.g., Pruppacher and Klett, 1997). Within parcel, cloud-

resolving and climate model microphysics schemes, ice properties are simplified in a variety of ways, generally based on some degree of observational guidance. Early observational studies using single-crystal measurement approaches commonly reported power-law relations between particle mass and a relevant particle dimension, such as column length or aggregate maximum dimension, generally valid over a relatively short range of dimensions measured for any particular crystal habit class (e.g., Locatelli and Hobbs, 1974). Later work identified the importance of projected area to fall speed, reported observation-based area-dimensional power laws for a few habits, and provided estimates for a number of others (e.g., Mitchell, 1996). Whereas the foregoing studies reported mass- and area-dimensional relations by habit, later studies attempted to use additional measurements such as circumference to obtain robust relationships that do not depend upon first assigning a habit (e.g., Baker and Lawson, 2006a; Schmitt and Heymsfield, 2010), an approach that is desirable in part owing to the fact that crystal habit is commonly irregular (e.g., Korolev et al., 2000). A convenient aspect of power-law relations, whether they are derived for one habit or a mixture, is their ease of analytical integration in parameterized microphysics schemes (e.g., Morrison et al., 2005).

In very detailed modeling studies of ice evolution, if habit geometry is well defined, precise calculations can be made for capacitance and other microphysical parameters (e.g., Hashino and Tripoli, 2011). However, even in a natural cloud system where nearly all crystals are in the same habit class, that habit may be characterized by chaotically polycrystalline shapes (e.g., Bailey and Hallett, 2002), as in the case of radiating plates seen during the Surface Heat Budget of the Arctic campaign (e.g., Fridlind et al., 2012), or may be subject to wide diversity of form, as in the case of dendrites ranging from plate-like to star-like seen during the Indirect and Semi-Direct Aerosol Campaign (e.g., Avramov et al., 2011). The fact that the majority of ice crystals in natural clouds are not generally pristine owing at least in part to the commonality of polycrystalline growth and the curving sides and edges caused by sublimation (e.g., Korolev et al., 1999) has been increasingly recognized in literature that addresses the consequences of morphological diversity for factors such as single-scattering properties (e.g., McFarquhar et al., 1999). Later laboratory and measurement analyses have specifically aimed to provide more generalized guidance on complex morphologies, offering revisions to earlier diagrams of habit as a function of temperature and supersaturation (e.g., Bailey and Hallett, 2002; Korolev and Isaac, 2003; Bailey and Hallett, 2009).

Currently, based on CPI imagery, automated identification of ice habit is relatively commonly reported (e.g., Lawson et al., 2006b; McFarquhar et al., 2007). However, analysis of quantitative single-crystal data on within-habit diversity to inform the representation of microphysical and radiative properties of ice for modeling studies of observed case studies (or, by extension, cloud system classes such as cirrus) remains nearly absent. The widespread occurrence of polycrystals and aggregates further complicates ice properties substantially. In relatively thick mixed-phase clouds, for instance, cycles of riming and vapor growth may result in a wide variety of plate-like fin structures grown on highly rimed substrates (e.g., Magono and Lee, 1966, R3c habit) as seen during the Mixed-Phase Arctic Cloud Experiment (Fridlind et al., 2007), creating crystal properties so diverse that it is essentially impossible to find quantitative, measurement-based guidance from analyses available in the literature to date.

Perhaps not yet as widely considered in models are the difficulties of consistently assigning ice crystal component aspect ratio, roundness, and microscale surface roughness for accurate calculation of radiative properties (e.g., van Diedenhoven et al., 2014a). Crystal extinction, absorption and emissivity are mostly determined by crystal mass and area (Fu, 1996; Fu et al., 1999; van Diedenhoven et al., 2014a). Although the general habit of ice crystals impacts their shortwave radiative properties, most important shape aspects for scattering properties are the aspect ratio of the crystal components and the degree of surface roughness (Iaquinta et al., 1995; Um and McFarquhar, 2007, 2009; Fu, 2007; Baran, 2009; van Diedenhoven et al., 2014a). When using mass- and area-dimensional relations as a foundation for ice properties in a model, as most commonly done, it is possible to assign a surface roughness and aspect ratio, and to calculate optical properties based on columns and plates that match ice volume, projected area and aspect ratio for any given ice class and size (e.g., Fu, 1996, 2007; van Diedenhoven et al., 2012). Guidance can be obtained from past studies of cirrus that quantify the variability of bullet arm aspect ratio, for instance (e.g., Iaquinta et al., 1995; Heymsfield and Iaquinta, 2000; Um and McFarquhar, 2007). However, the aspect ratio of whole crytals and their crystalline elements are relatively scarcely reported and analyzed for natural ice crystals (e.g., Korolev and Isaac, 2003; Garrett et al., 2012; Um et al., 2015), making necessary some relatively poor approximations for specific natural conditions that may be encountered in the field (e.g., Fridlind et al., 2012). Finally, for a size-resolved microphysics scheme, obtaining continuity of ice particle properties over the full size range required to represent relevant cloud microphysics generally requires awkward concatenation of aspect ratio-, mass- and area-dimensional relations relevant for limited size ranges (e.g., van Diedenhoven et al., 2012; Sölch and Kärcher, 2010), which can easily lead to unphysical discontinuities in derived quantities such as fall speed or capacitance. Erfani and Mitchell (2016) recently provided polynomial mass- and area-dimensional relations that surmount lack of continuity and simplify to analytically integrable power laws that closely approximate the full solution over a local size range.

Here we analyze single-crystal ice crystal field data with the primary objective of deriving physically continuous ice

microphysical *and* optical properties over the size range required (1–3000 $\mu$m). As a well-defined starting place, and a foundation for large-eddy simulations, we focus narrowly on the morphological properties of a well-developed midlatitude synoptic cirrus case study, taking advantage of an existing extended analysis of single-crystal images (Um et al., 2015). Because the most accurate representation of cirrus optical properties requires consideration of polycrystal element aspect ratios (e.g., van Diedenhoven et al., 2014a), which are commonly a function of particle size in observations, the polycrystal elements are adopted as the foundation for treating mass and projected area rather than vice versa (as required if area- and mass-dimensional relationships are instead adopted as the foundation, as most commonly done); a similar approach was taken by Heymsfield and Iaquinta (2000) for the purpose of deriving physically based expressions for cirrus crystal terminal velocities, such as bullet rosettes with varying numbers of arms. Parcel simulations are used to compare the ice properties derived in this work with ice properties available in existing literature that have been used in large-eddy simulations of cirrus with size-resolved microphysics (Sölch and Kärcher, 2010). Because the derivations here are based on crystal component geometries and do not yield continuous analytic relationships, equations are provided in Appendix A and derived ice properties are provided for download as the Supplement.

## 2   Observations

In situ observations are analyzed from a well-sampled cirrus system observed during 1 April (flight B) and 2 April (flight A) during the 2010 Small Particles in Cirrus (SPARTICUS) field campaign (Mace et al., 2009). Based on an extensive analysis of atmospheric states during SPARTICUS, Muhlbauer et al. (2014) classified the 1–2 April conditions as ridge-crest cirrus (Fig. 1). Relative to the other non-convective cirrus states identified during SPARTICUS, ridge-crest cirrus were characterized by formation within the coldest environments at cloud top, within considerable ice supersaturation, and were statistically associated with the highest ice crystal number concentrations and lowest ice water contents.

Previous studies using SPARTICUS data can be considered in at least five general categories: characterization of the environmental properties observed (e.g., Muhlbauer et al., 2014), characterization of the ice crystal morphology or size distribution characteristics observed (e.g., Mishra et al., 2014; Um et al., 2015; Jackson et al., 2015), cirrus cloud process modeling studies (e.g., Jensen et al., 2013; Muhlbauer et al., 2015), evaluation of satellite retrievals (e.g., Deng et al., 2013), and evaluation of climate model cirrus properties (e.g, Wang et al., 2014). The work here is in the second category, and is based primarily on single-crystal ice crystal properties using data obtained from a CPI probe on the

Stratton Park Engineering Company (SPEC) Inc. Learjet 25 aircraft. The ice crystals imaged by the CPI are first classified by habit using the scheme described by Um and McFarquhar (2009). Images classified as bullet rosettes and aggregates of bullet rosettes are then further analyzed using output from the recently developed Ice Crystal Ruler (ICR) software (Um et al., 2015) to obtain the imaged width and length of each branch.

To provide context for parcel simulations, we also use Learjet ice particle size distributions derived from a 2D Stereo Probe (2DS) equipped with tips that reduce effects of shattering (Lawson, 2011) and analyzed as reported by Jackson et al. (2015), together with in-cloud vertical wind speed retrievals from profiling Doppler radar measurements (Kalesse and Kollias, 2013).

## 3   Model description

The overall objective of this study is to use analyzed CPI image data to derive consistent representations of ice physical and optical properties for a size-resolved ice microphysics scheme, and to compare results with existing literature. The target microphysics scheme is based on the Community Aerosol-Radiation-Microphysics Application (CARMA) code (Jensen et al., 1998; Ackerman et al., 1995). The CARMA model allows selection of an arbitrary number of mass bins to represent the size distributions of an arbitrary number of aerosol and ice classes. Within each ice class, the mass in each bin is a fixed multiple of the mass in the preceding bin.

In this work the ice crystal properties in each ice mass bin are represented using the approach developed by Böhm (1989, 1992b, c, a, 1994, 1999, 2004), as previously applied to represent the ice crystals in mixed-phase stratus in Avramov et al. (2011, dendrites and their aggregates) and Fridlind et al. (2012, radiating plates). The Böhm scheme provides an integrated treatment of terminal fall speeds and collision efficiencies for non-spherical ice that is based not on specification of a particular habit but rather on four properties that are quantitatively defined for both pristine and non-pristine shapes: particle mass $m$, a characteristic maximum dimension $D$ and projected area $A$, and aspect ratio $\alpha$. The foundational physical quantity of this parameterization is fall speed, so the characteristic quantities $A$, $D$ and $\alpha$ are best defined by fall orientation, which can perhaps most simply be considered as the maximum projected area (which determines the fall orientation), the maximum dimension of a circumscribed circle around that projected area, and the aspect ratio of thickness normal to the fall orientation to that maximum dimension (cf. Böhm, 1989). Bodily aspect ratio $\alpha$ is defined as 1 for ice crystals without a preferential fall orientation (e.g., bullet rosettes), less than 1 for oblate bodies (e.g., plates), and greater than 1 for prolate bodies (e.g., columns). Throughout this work, $\alpha$ is fixed at 1 based on the geome-

tries discussed below, and $D$ and $A$ by extension assumed equal to randomly oriented maximum dimension ($D_{max}$) and randomly oriented projected area ($A_p$).

In each ice mass bin, quantities that are not considered in the Böhm scheme but that should ideally be specified in an integrated manner are capacitance and radiative scattering and absorption coefficients. For a given crystal, the capacitance can be either specified from the literature in the case of a pristine habit or else estimated from prolate or oblate spheroids (Pruppacher and Klett, 1997, their Eqns. 13-78 and 13-79). Here we take the former approach for bullet rosettes and their aggregates and polycrystals analyzed below: given bullet arm length $L$ and arm width $W$ (twice the hexagon side length), we specify $D_{max}$-normalized capacitance ($C$) as $0.4(L/W)^{0.25}$ based on the fit to calculations for six-arm rosettes by Westbrook et al. (2008).

Scattering and absorption properties assuming randomly oriented ice crystals in each mass bin are computed following van Diedenhoven et al. (2014a), which, in addition to $m$ and $A_p$, also requires specification of elemental aspect ratio ($\alpha_e$) and a microscale surface roughness or crystal distortion ($\delta$), as defined by Macke et al. (1996). Here $\alpha_e$ required for the optical properties is identical to the bodily aspect ratio $\alpha$ in the case of a single-component crystal (e.g., plate or column), but for a polycrystal such as a bullet rosette $\alpha_e$ is the aspect ratio of constituent arms or other component crystals (cf. Fu, 2007). In this work $\alpha_e$ values are derived from ICR measurements where possible. Additional details are given in Section 5.4.

Parcel simulations are used to test ice properties in a simplified framework, following Ackerman et al. (2015), prior to use in computationally expensive 3D large-eddy simulations in future work. All simulations include adiabatic expansion, aerosol homogeneous freezing, diffusional growth of ice crystals, and latent heating. Heterogeneous freezing is neglected. Parcels are initialized at 340 mb, 233 K, and 80% relative humidity. Saturation vapor pressures are related to water vapor mixing ratio following Murphy and Koop (2005). Each simulation is assigned a fixed updraft speed ($w$) of 0.01–1 m s$^{-1}$. Parcel expansion is treated by assuming dry adiabatic ascent and iterating three times on parcel air pressure, temperature, and density assuming hydrostatic conditions and using the ideal gas law. Latent heat is computed in accord with diffusional growth of the ice. A default time step ($\Delta t$) of 1 s is variably reduced to a minimum value of 0.1 s, which is reached when fast processes such as aerosol freezing are active, and parcel height is incremented by $w\Delta t$ each time step. Ice sedimentation, when included, assumes a vertical length scale of 100 m as in Kay and Wood (2008). Gravitational collection is neglected. We use the Koop et al. (2000) parameterization for aerosol freezing, including the Kelvin effect on surface vapor pressure, and assume that aerosol are at equilibrium with atmospheric water vapor. Aerosol are initialized with a concentration of 200 cm$^{-3}$ lognormally distributed with geometric mean diameter 0.04 $\mu$m and geometric standard deviation 2.3 as in Lin et al. (2002), except that composition is assumed to be ammonium bisulfate. We assume a fixed ice accommodation coefficient of 1, which is within the range of recent laboratory measurements (Skrotzki et al., 2013), and account for Knudsen-number-dependent gas kinetic effects (cf. Zhang and Harrington, 2015). Growth across mass bins is treated with the piecewise parabolic method of Colella and Woodward (1984). Simulations use 50 bins with a mass ratio of 1.65 from one bin to the next, starting with $D_{max}$ of 2 $\mu$m, suitable for use in 3D large-eddy simulations.

## 4   Derivation of ice single-crystal properties

Considering all CPI images collected during the 1–2 April flights, automated analysis places roughly half of all ice crystals in the small quasi-sphere category, and remaining crystals are primarily unclassified (Fig. 2). However, considering only ice crystals with $D_{max}$ greater than 100 $\mu$m, bullet rosettes emerge as the most common classified habit. Subjective examination of images suggests that bullet rosettes are the dominant habit in the coldest crystal growth regions with significant ice water content (Fig. 3), as discussed further below. We therefore begin with an analysis of ICR measurements of bullet rosette arm lengths and widths, which are suitable to describe the physical and optical properties for a cloud composed entirely of growing rosettes.

### 4.1   Bullet model

For each bullet rosette measured with the ICR software, Fig. 4a shows mean branch length versus measured $D_{max}$. Since branches that are not aligned with the viewing plane are foreshortened, we take $L$ as the average of all measured branch lengths minus half of randomly oriented projected end plate diameter, multiplied by a factor of $4/\pi$ to account for random orientation to first order (see Appendix A1). The relationship of $L$ to $D_{max}$ is reasonably fit by a line passing through the origin.

For the same crystals, Fig. 4b and Fig. 4c show mean $W$, and the ratio $L/W = \alpha_e$, respectively. To account for random orientation, $W$ is taken as the average of all measured branch widths divided by a factor of $(1+\sqrt{3}/2)/2 = 0.933$, which is the ratio of the arithmetic mean of mimimum and maximum branch projected widths to the maximum (equivalent to the ratio that would be found if measurements of projected width were made for a sufficiently large number of orientations of a bullet arm of known $W$). Both mean and median number of branches is six (out of four to ten measured), consistent with recent analyses from tropical and Arctic field campaigns (Um et al., 2015). Rosettes with more branches are seen to have systematically smaller $W$ and larger $\alpha_e$, consistent with competition for vapor during growth. However, a simple least squares fit of $W$ to $D_{max}$ gives $W > D_{max}$ when extrapolated

to small crystal size, which is not physical; unfortunately, measurements are not available to provide guidance at such sizes.

Because we seek a continuous description of ice properties across all sizes, here we take the approach of adopting a physical model of crystal geometry to extrapolate measured properties smoothly to sizes smaller than measured. A similar approach was taken by Heymsfield and Iaquinta (2000) to improve calculated cirrus crystal fall speeds over those obtained from independently derived mass- and area-dimensional relations. We first assume branch width for rosettes consistent with the six-rosette model considered in Westbrook et al. (2008), but using a fixed angle of 44° between opposing edges of the hexagonal pyramids that cap each branch (sensitivity of results to choice of cap angle is discussed at the end of Appendix A1). Selecting a fixed angle and using the linearly fit branch width at all sizes allows determination of the cap contribution to $L$; $L$ is found to consist entirely of a truncated cap at the smallest sizes and corresponding $W$ is taken as the truncated cap base width. This model results in the line slope discontinuity seen in Fig. 4b, and resolves at least gross discrepancy of $W > D_{max}$.

Fig. 4c shows that adopting this bullet model results in a smooth increase in branch aspect ratio $\alpha_e = L/W$ from smallest to largest sizes, suitable as a basis for calculating optical properties. $L/W$ is constant at the smallest sizes, where only the cap contributes and both $W$ and $L$ are varying at the same relative rate. The range of aspect ratios measured (2–6) and the fitted trend from near-unity at the smallest sizes to roughly 5 at the largest sizes is consistent with several past studies (cf. Heymsfield and Iaquinta, 2000). As shown, the relationship of branch aspect ratio to $D_{max}$ also agrees with that used by Mitchell (1994) in derivation of the mass-dimensional relation for $D_{max} > 100\,\mu$m listed in Table 1 and discussed further below.

The Westbrook et al. (2008) model assumes that all bullets are at 90° angles to one another, giving true maximum dimension of $2L$, which is ~40% greater than $D_{max}$ shown in Fig. 4a. Measured $D_{max}$ being a randomly oriented value can account for less than 30% discrepancy. Another source of difference is the commonly seen deviations of arm locations from 90° separations, which can only decrease $D_{max}$ from $2L$. Since a more quantitative explanation is beyond the scope of this initial study, we adopt the randomly oriented $D_{max}$ as our only defined maximum dimension, an assumption that has also been made in past studies using two-dimensional images (e.g., Heymsfield et al., 2002; Baker and Lawson, 2006a).

The bullet model described above now allows calculation of crystal surface area ($A_s$) and $m$ (see Appendix A1 for details). To calculate $m$ from the geometrical dimensions, we assume ice bulk density ($\rho_i$) of 0.917 g cm$^{-3}$; any bullet arm hollows are neglected here owing to lack of quantitative guidance, as discussed further below. Calculated $m$ and $A_p$ of a six-branch rosette are seen to reasonably represent the scatter of individual crystal properties (solid lines in Fig. 4e and 4f). The ratio of measured $A_p$ to calculated $A_s$ is found to be about 0.11 (Fig. 4d), smaller for these concave particles than the $A_p/A_s$ of 0.25 for convex shapes, consistent with theoretical results (Vouk, 1948) and reasonably independent of $D_{max}$ across measured sizes.

In Fig. 4e derived $m(D_{max})$ is compared with power-law relations from previous literature that have been used in similar bin microphysical schemes (Table 1). To our knowledge, only one unpublished data set has provided direct measurements of bullet rosette mass, consisting of 45 crystals with a range of 2–5 arms as reported by Heymsfield et al. (2002), but that data set is not the basis of commonly used relations. As used in Sölch and Kärcher (2010), for instance, the Heymsfield et al. (2002) relation is based on calculation of effective density from a combination of ice water content and particle size distribution measurements; coefficients in Table 1 (cf. Sölch and Kärcher, 2010) are calculated from their Equation 22, based on crystals with $D_{max}$ of 200—2000 $\mu$m fit in their Fig. 15. As also used in Sölch and Kärcher (2010), Mitchell (1994) combined crystal volume expressions with size-dependent bulk densities of ~0.78 g cm$^{-3}$ to obtain an $m - D_{max}$ relation for crystals with $D_{max}$ of 200–1000 $\mu$m (Table 1 values are taken from their equation 32); for crystals smaller than 100 $\mu$m, Mitchell et al. (1996) proposed a mass-dimensional relation using ad hoc estimates of crystal mass (Table 1 values are taken from their Table 3).

The difference between our calculated $m(D_{max})$ and that from Mitchell (1994) is roughly a factor of four at measured crystal sizes, which results in a similar discrepancy in fall speeds and effective diameters, as shown below. We can attribute lower $m$ in Mitchell (1994) to four factors: (i) $L$ is substantially shorter based on the approximation $D_{max} = 2L$ (cf. Iaquinta et al., 1995, including assumed trilateral pyramidal end following their Fig. 1), (ii) $W$ is substantially thinner based on earlier cited literature that relates $W$ to $L$ and by extension $D_{max}/2$ (see Fig. 4b), (iii) five branches are assigned instead of six found here, and (iv) $\rho_i$ is ~0.78 g cm$^{-3}$ instead of 0.917 assumed here. All else being equal, increasing their branch number and $\rho_i$ would together increase Mitchell (1994) $m$ by only about 40%, but $m$ scales roughly linearly with $L$ and geometrically with $W$. The trilateral pyramid ends taken from Iaquinta et al. (1995) would result in slightly greater $m$ than ours, all else being equal. The close agreement between our arm aspect ratio $L/W$ and that following Mitchell (1994), available for $D_{max} > 100\,\mu$m (Fig. 4c), suggests that differences in $m$ are primarily attributable to differing approaches to defining $D_{max}$. However, we are unable to quantitatively confirm that because randomly oriented maximum dimension cannot be calculated analytically for either the idealized geometries derived here or for CPI images of natural crystals.

Our calculated $m(D_{max})$ is also nearly a factor of two greater than that from Heymsfield et al. (2002) for ice particle ensembles (all habits, dominated by bullet rosettes) mea-

sured over the same Oklahoma location. In Heymsfield et al. (2002), $D_{max}$ is taken from 2D Cloud and Precipitation Probe measurements and $m$ is derived from coincident ice water content measurements from a Counterflow Virtual Impactor (CVI) via a linear fit of effective particle density ($\rho_e$, the density of a sphere with diameter $D_{max}$) to $D_{max}$. Whereas our approach is subject to uncertainty in ICR measurements and assumed $\rho_i$, the Heymsfield et al. (2002) approach is subject to uncertainty in the measurement of ice particle size distribution, uncertainty in the measurement of ice water content, and the importance of any deviations of the particle ensemble from bullet rosettes. Uncertainty in CVI probe measurements are reported to be 10% for ice water contents larger than 0.2 g m$^{-3}$ (Twohy et al., 1997), but the bin-wise uncertainty in particle size distribution measurements are generally unquantified; we consider it beyond the scope of this study to undertake the detailed analysis required to resolve such differences. Although it is not used in the $m - D_{max}$ relationship adopted by Sölch and Kärcher (2010) and listed in Table 1, Heymsfield et al. (2002) also derive a typical $\rho_i$ of 0.82±0.06 g cm$^{-3}$ for bullet rosettes based on independent photographic evidence for hollow bullet rosette arm ends; we make no such reduction here, as discussed above, and doing so is not a dominant cause of the differences in $m$.

Whereas our calculated $m(D_{max})$ is substantially greater than that previously used in studies with size-resolved microphysics, our measured $A_p(D_{max})$ is similar or smaller. The relationship of $A_p$ and $D_{max}$ derived by Mitchell et al. (1996, their Table 1) independently from $m(D_{max})$ for five-branched bullet rosettes in a manner similar to that here, is nearly identical to ours (cf. Fig. 4e). The less widely available $A_p$–$D_{max}$ relations are surprisingly more difficult to trace, considering that they can be more directly derived from CPI images, and we are unable to identify the observational sources of the relations reported in Sölch and Kärcher (2010), which are cited from but not apparent in Heymsfield et al. (2002).

Figure 5 allows a closer examination of the extrapolation from manually measured rosette properties ($D_{max} > 200\ \mu m$) to smaller sizes using our bullet model, and shows comparisons to additional published fits. From in situ measurements of total ice water content and ice crystal size distribution and shape obtained from a 2DS probe in mid-latitude cirrus, Cotton et al. (2012) derived a mean $\rho_e$ of 0.7 g cm$^{-3}$ below a threshold size of 70 $\mu m$ and a power law decrease of density to 0.5 g cm$^{-3}$ at roughly 100 $\mu m$ and 0.05 g cm$^{-3}$ at roughly 1000 $\mu m$. The mean $\rho_e$ derived here happens to exhibit a similar behavior (Fig. 5a), where the discontinuity using our bullet model represents the transition to truncated branch caps. Erfani and Mitchell (2016) derived polynomial $m - D_{max}$ relations for synoptic cirrus clouds warmer than $-40°C$ from single-particle measurements of $m$, $D_{max}$ and $A_p$ obtained during the 1985–1987 Sierra Cooperative Pilot Project (SCPP) (Mitchell et al., 1990), or by applying a habit-independent $m - A_p$ relation derived from the SCPP data set (Baker and Lawson, 2006a), shown in Figure 4e, to

2DS measurements obtained during 13 SPARTICUS flights (at colder temperatures). Although the SCPP data set does not contain bullet rosettes or spatial crystals (Baker and Lawson, 2006a), Lawson et al. (2010) report that ice water content derived by applying that habit-independent $m - A_p$ relation to a combination of tropical anvil and synoptic cirrus measurements agreed with CVI measurements to within 20%. At $D_{max} < 100\ \mu m$, Erfani and Mitchell (2016) $m$ values were calculated from CPI measurements of $A_p$ and $\alpha$ assuming hexagonal column geometry (cf. Erfani and Mitchell, 2016, their Appendix B), and effective densities are similar to those derived here. At larger sizes and especially colder temperatures, Erfani and Mitchell (2016) effective densities are smaller than derived here, consistent with the (Baker and Lawson, 2006a) $m - A_p$ relation giving lower per-particle $m$ than derived here.

Although $m$ cannot be calculated in this study for bullet rosettes that are not measurable with the ICR software or for crystals with unclassified habit, $A_p$ is reported for all imaged crystals and can be directly compared with the bullet model. Analogous to $\rho_e$ but dimensionless, the measured ratio of $A_p$ to that of a sphere with diameter $D_{max}$ can also be compared with the bullet model. Figure 5b shows that literature power law relations can become unphysical for the smallest particle sizes (projected areas greater than for a sphere of diameter $D_{max}$); to correct the greatest deviations for the purposes of parcel calculations below, we adopt a constant ratio of $A_p$ to sphere projected area where $D_{max} < 100\ \mu m$ when using Mitchell (1994) relations. When considering all rosettes automatically identified (not all of which were measurable using the Ice Crystal Ruler), the bullet model $A_p(D_{max})$ agrees quite well with median measurements and with $m - A_p$ relations for bullet rosettes and budding bullet rosettes from Lawson et al. (2006a) (Fig. 5c). However, when considering all crystals (Fig. 5d), there is a wider range of $A_p(D_{max})$ and the bullet model underestimates median $A_p(D_{max})$, as addressed further below; the Erfani and Mitchell (2016) polynomial $m - A_p$ fit from the SCPP data set (their warmest-temperature fit) and the Heymsfield et al. (2002) $m - A_p$ power law agree best with the full data set where $D_{max} > 100\ \mu m$.

## 4.2 Bucky ball model

To consider uncertainty in the geometry of the smallest crystals, we next consider an alternative proposed model for early bullet rosette shape: budding Bucky balls (Um and McFarquhar, 2011). The so-called budding rosette shape has been observed in laboratory grown ice and ice-analog crystals (Ulanowski et al., 2006; Bailey and Hallett, 2009), and the CPI does not have the resolution necessary to distinguish such a shape from the bullet model geometry assumed above. Here we approximate the Bucky ball core as a sphere of diameter 10 $\mu m$ and then assume that arms emerge with initial width 4 $\mu m$. If we assume that $L$ falls linearly to zero at $D_{max}$

equal to the core dimension (Fig. 6a) and $W$ correspondingly falls linearly to its minimum initial width (Fig. 6b), then branch $\alpha_e$ is relatively constant near the mean observed (Fig. 6c). $A_s$ and $m$ can now be calculated using this Bucky ball model, except that $A_p$ of the smallest crystals must be interpolated to bridge the geometry of a sphere ($D_{\max} <$ core diameter) and that of a rosette; to do this, we calculate a mass-weighted sum of $A_p$ obtained from the linear relation in Fig. 4d and that of a sphere (see Appendix A2). Thus, as $m$ converges to that of a sphere, so does $A_p$. Results are similar to those of the bullet model at larger particle sizes (Fig. 6d-f), with $m(D_{\max})$ still larger than previous estimates and $A_p(D_{\max})$ still similar or smaller.

However, using this simplified Bucky ball model, a developing six-arm rosette has a systematically smaller $\rho_e$ and $A_p$ than it did with the bullet model (Fig. 7). Although this particular version of a Bucky ball model, with only six arms even at small sizes, gives substantially smaller $A_p(D_{\max})$ than measured for automatically classified rosettes at small $D_{\max}$ (Fig. 7c), it does serve to provide a quite close match to the minimum area relative to that of a sphere over the full particle data set (Fig. 7d), and is therefore included in parcel calculations below. In reality it seems likely that not all budding arms grow evenly. For instance, Um and McFarquhar (2011) propose a Bucky ball model with 32 regular and irregular hexagonal arms, one growing from each of the ball's 20 hexagonal and 12 pentagonal planes. From this study, it is apparent that only up to about 12 arms commonly reach substantial lengths, and most commonly only six such arms are seen. Faced with the problem of how to introduce geometry that smoothly transitions from an unknown larger number of sub-100-$\mu$m arms to roughly six arms at larger sizes with no quantitative basis for how to introduce such added complexity here, we have simply assumed six arms throughout.

### 4.3 Aggregate model

We return now to the distribution of habits during the April 1–2 flights, and consider the properties of crystals in the observed cirrus deck that are not identified as bullet rosettes. The rosettes are most common in the upper cloud regions at temperatures colder than $-40°C$ (Fig. 8), consistent with previous findings that rosette shapes in the temperature range $-40$ to $-55°C$ are mostly pristine (Lawson et al., 2006a). In this case, at slightly warmer temperatures, aggregates of bullet rosettes become most common. Using ICR measurements for aggregates of bullet rosettes, it is straightforward to extend the bullet model to rosette aggregates (Fig. 9, see Appendix A3), where the mean and median branch numbers are found to be 12 per aggregate, consistent with aggregation of two typical bullet rosettes. Compared to single rosettes, aggregate properties are generally similar to those of single rosettes except shifted in size to a larger maximum dimension. We do not dwell here on the properties at the smallest sizes since aggregates are born from fully

formed bullet rosettes and this study is focused on crystal growth (neglecting sublimation).

However, aggregates of pristine rosettes also represent a small fraction of ice crystals observed in this case, at least on a number basis. CPI images show that some rosettes reach a plate growth regime (Fig. 10), a phenomenon well documented in previous cirrus field observations and laboratory measurements (Bailey and Hallett, 2009). In the lower cloud regions at temperatures warmer than $-40°C$, modified bullets have been described as mostly "platelike polycrystals, mixed-habit rosettes, and rosettes with side planes" (Lawson et al., 2006a), where side plane growth on columns may be attributable to facet instability on prism faces (Bacon et al., 2003).

### 4.4 Polycrystal model

For the purposes of considering how plate-like growth impacts rosette single-crystal properties, it is notable from the SPARTICUS images in this case that radiating side plane elements appear to increasingly fill the space between the arms of rosettes and rosette aggregates, giving the impression of cobwebs that lead to blocky ice particle shapes (e.g., Fig. 10). In such a process, particle $m$ could increase without rapid expansion of particle $D_{\max}$. Such a tendency for crystals to become less florid may be related to the finding of side plane growth on rosettes in the laboratory exclusively originating from the rosette center, consistent with an important role for defect and dislocation sites (Baker and Lawson, 2006b). Toward cloud base, sublimation then increasingly rounds crystal edges (Fig. 11). Rosettes that did not enter a side plane growth stage appear now with rounded arms that can still be counted, whereas rosettes that did experience substantial side plane growth emerge from sublimation zones as relatively large quasi-spheres, which appear as a non-negligible percentage of large particle habit; the existence of such large quasi-spheres would be otherwise difficult to explain. The smallest sublimated crystals appear occasionally as sintered chains.

We next consider an approximate model for the physical and optical properties of these more common, irregular crystals. In the data set examined here, we are unable to find a consistent increase in projected area ratio with increasing temperature that would be expected if rosettes are modified by side plane growth during sedimentation from colder to warmer temperatures, but we do find that unclassified crystals at all temperatures exhibit consistently larger area ratios than rosette crystals (Fig. 12). To account for rosette shape evolution in a manner amenable to calculation of radiative and microphysical properties at least for growing crystals, we attempt to coarsely estimate the side plane mass added to pristine rosettes and its associated elemental aspect ratio as follows.

We first calculate the additional projected area that can be attributed to side plane growth. Considering all unclassified

crystals, a fit of measured $A_p$ to calculated bullet surface area (based on measured maximum dimension and assuming a bullet model rosette with six arms) yields a slope of 0.15 (Fig. 13a), which is larger than the slope of 0.11 found using the bullet model for measured rosettes, consistent with greater area ratios for unclassified crystals. If we make the ad hoc assumption that the relationship of surface area to projected area is close to that for bullet rosettes, we can attribute the surface area beyond that of the bullet model to plates. If we make the ad hoc assumption that a plate-like side plane grows on each of six arms and neglect plate thickness, the plate or side plane surface area can be considered as the sum of hexagonal faces of the six plates, and the plate diameter can be calculated. If we further relate plate thickness to plate diameter as described in Appendix A4, then mass can now be calculated as the sum of bullet and plate contributions for a typical particle (e.g., Fig. 13c, solid line). For this crude representation of plate-like growth on the bullet model, the calculated crystal properties agree reasonably well with Cotton et al. (2012) effective density in the limit of small $D_{\max}$ (Fig. 13e) and with the area ratio as a function of $D_{\max}$ over all unclassified crystals (Fig. 13f) if the following choices are made: the cap angle $\beta$ is increased to 25°, plates are assumed present only where branches extend beyond truncated caps, and the plate surface area is assumed to increase inverse exponentially to its terminal value with a length scale equal to the diameter at $L$ greater than $L_c$ (see Appendix A4 for details). Where $D_{\max} > 100\ \mu$m, the resulting polycrystal model $A_p$ agrees closely with the Erfani and Mitchell (2016) fit for warmest-temperature synoptic cirrus, but resulting $m$ is now also correspondingly greater than and further from Erfani and Mitchell (2016) than in the bullet model (cf. Fig. 5a).

The foregoing results for this polycrystal model are dependent upon the underlying bullet model assumed, the assumed ratio of $A_s$ to $A_p$, and the assumed plate or side plane geometry, for which no quantitative guidance exists in the current data set. This polycrystal model is intended only as a relatively simple example of ice properties that is guided by available observations and allows calculation of internally consistent physical and radiative properties in a continuous fashion over all crystal sizes that need to be represented in our microphysics model. In order to evaluate the need for further consideration of ice properties in greater detail, we next consider parcel simulations to evaluate the influence of ice models on predicted size distributions and optical properties.

## 5 Model results

### 5.1 Fall speed and capacitance

Figure 14 shows a point calculation of fall speeds ($v_f$) at 350 mb and 233 K for comparison with Sölch and Kärcher (2010, their Fig. A1). Our bullet model gives $v_f(D_{\max})$ values that are more than a factor of 1.5–2 greater than derived from Mitchell and Heymsfield ice properties and used by Sölch and Kärcher (2010) in a microphysics model similar to ours. Increasing crystal $m$ calculated from the literature by a factor of 0.917/0.78 can account for relatively little of the difference (not shown), indicating that the main differences are attributable to crystal geometries. In the case of Mitchell properties, as discussed above, the main difference may be traceable to differing $D_{\max}$ definition used to calculate $m$, whereas $A_p(D_{\max})$ is very similar. In the case of Heymsfield properties, $m(D_{\max})$ is closer to ours but $A_p(D_{\max})$ is also larger, a factor that should be relatively more easily resolved in future studies since both $A_p$ and $D_{\max}$ can be directly measured. As shown in Fig. 5c, for instance, an $m - A_p$ relation from earlier midlatitude cirrus measurements (Lawson et al., 2006a) agrees well with SPARTICUS rosette measurements and with our bullet model. At the warmest temperatures considered by Erfani and Mitchell (2016), $A_p(D_{\max})$ is similar to or even greater than the bullet or polycrystal models but substantially lower $m(D_{\max})$ leads to substantially lower $v_f(D_{\max})$; the Erfani and Mitchell (2016) trend toward greater decrease in $m$ than $A_p$ with decreasing temperature leads to increasing divergence between the models derived here and their results.

Given literature ice properties, using our model to calculate crystal fall speed as detailed in Avramov et al. (2011) results in $v_f$ values that appear similar to those of Sölch and Kärcher (2010) and are also within roughly 10% of those calculated using the method described in Heymsfield and Westbrook (2010) (not shown). However, our aggregate model gives fall speeds roughly one-third reduced from similar-sized bullet model ice, which is a substantially larger difference than that using Heymsfield ice properties for aggregates and their rosettes shown in Sölch and Kärcher (2010). We can trace this greater difference in part to substantially larger $m(D_{\max})$ derived here, as shown above. Overall, we conclude from comparison of our results with those of Sölch and Kärcher (2010) that the precise method of calculating $v_f$ as a function of $m$ and $A_p$ appears to be responsible for relatively little spread, but differences in ice properties themselves ($m$ and $A_p$) introduce $v_f(D_{\max})$ differences that are substantially larger than expected, as discussed further below.

Owing to the dependence of parameterized capacitance on bullet arm aspect ratio alone (see Section 3), capacitance differences are nearly negligible for crystals larger than ~400 $\mu$m across all bullet models derived here, in sharp contrast to factor of two differences in fall speed at such sizes. Because assumed or derived bullet arm aspect ratios vary most where $D_{\max}$ is less than 300 $\mu$m, capacitance differences up to roughly 25% are most pronounced at those sizes. Although aspect ratios used in derivation of the Mitchell ice properties are similar to ours where $D_{\max} > 100\ \mu$m (see Fig. 6), no such aspect ratios are provided for smaller Mitchell crystals or for Heymsfield ice properties. For parcel calculations, we therefore adopt a $C$ value of 0.25 derived

for aggregates (Westbrook et al., 2008), taken here as representative of polycrystals with unspecified aspect ratios. A similar assumption would be required for Erfani and Mitchell (2016) ice properties; since parcel simulations are also not configured for changes in ice crystal properties during a single simulation, we omit Erfani and Mitchell (2016) ice properties from the remaining calculations.

## 5.2 Parcel simulations without sedimentation

To grossly evaluate the potential effect of different model ice properties on ice crystal nucleation and growth, we first consider parcel simulations without the complication of sedimentation. Since aggregation is neglected, aggregate ice properties are not considered. As described in Section 3, parcels begin at 233 K ($-40°$C), 340 mb, and 80% relative humidity. Vertical wind speed ($w$) is fixed at 0.01, 0.1 or 1 m s$^{-1}$, within the range of millimeter cloud radar retrievals of in-cloud $w$ from the beginning of the first flight examined here to the end of the second flight (Fig. 16). Aside we note that a parcel simulation is not a realistic rendition of natural cirrus cloud evolution, which is characterized by extensive growth and sublimation during particle sedimentation. But a similar framework has been used to test cirrus models (Lin et al., 2002), and in this case it allows a simple comparison of particle growth to sizes that span the range observed during SPARTICUS, as discussed further below.

Figure 17 shows the ice particle size distribution (PSD) for each simulation at $-55°$C, which is close to the average of in-cloud temperatures observed during the 1–2 April flights. Figs. 18–21 show the ice crystal number concentration ($N_i$), number-weighted mean diameter ($D_i$), total projected area ($A_i$), and relative dispersion ($\nu$), respectively, as a function of parcel ascent distance. In the absence of sedimentation, ice mass is essentially distributed across differing crystal sizes depending upon $w$ and $\rho_e$. The magnitude of $w$ primarily determines $N_i$: when $w$ is strongest, vapor growth competes least with nucleation, resulting in greatest $N_i$ (Fig. 18a). Nucleated number concentrations range from several per liter when $w$ is 0.01 m s$^{-1}$ to several per cubic centimeter when $w$ is 1 m s$^{-1}$, consistent with past studies (e.g., Lin et al., 2002). The Heymsfield ice properties give roughly a doubling of $N_i$ relative to other ice properties, owing to the densest small ice accompanied by a fixed capacitance for non-spheres (in the absence of obvious means of transitioning capacitance from spheres to non-spheres).

The strongest $w$ and associated fastest aerosol freezing, which leads to largest $N_i$, leads to the correspondingly smallest $D_i$ (Fig. 19a) and the greatest $A_i$ (Fig. 20a). Where $N_i$ is insensitive to ice properties, the sensitivity of $D_i$ and $A_i$ to ice properties at a given $w$ can be seen as simply scaling inversely with effective density and area per unit mass, respectively. Ice properties assumptions lead to roughly a factor of 2 range of $D_i$ at lowest $w$ and nearly a factor of 4 range of $D_i$ at highest $w$. Whereas $D_i$ is variable with ice properties,

$A_i$ at all $w$ falls into two groups: Mitchell and Heymsfield properties, with relatively large $A_i$, and all other ice properties including spheres and our models derived here, with $A_i$ systematically smaller by roughly a factor of 3 at all $w$. Dispersion ($\nu$) exhibits up to factors of 2–3 difference (Fig. 21e). The ice properties associated with the lowest effective density (Mitchell) have greatest $D_i$ and $\nu$. However, at lowest $w$ the Bucky ball model exhibits substantially greater $D_i$ but similar $\nu$ as spheres, which can be attributed to a weak dependence of $\rho_e$ on $D_{max}$ at $D_{max} > 100~\mu$m that is more similar to spheres than other ice models (Fig. 7).

In summary, in the simple case of a non-sedimenting parcel, differing ice property assumptions lead to a factor of 2 difference in $N_i$ and factor of 3 in $A_i$. Up to a factor of 2 increase in $\nu$ is also induced by ice properties that exhibit a trend in $\rho_e$ across the relevant size distribution relative to ice properties with constant $\rho_e$. Differences in $\rho_e$ across ice properties considered here (regardless of trend) also lead to factors of 2–3 difference in $D_i$.

## 5.3 Parcel simulations with sedimentation

When sedimentation is included with an assumed parcel depth of 100 m following Kay and Wood (2008), results are largely unchanged at the strongest $w$ since $v_f \ll w$ (cf. Fig. 14); the only notable change is roughly a factor of two reduction in $N_i$ by $-55°$C, seen primarily as a uniform downward shift of the PSDs between Fig. 17a and Fig. 17b. However, at $w \ll 1$ m s$^{-1}$, the parcel behavior changes rather dramatically because sedimentation reduces surface area sufficiently to allow aerosol freezing events repeatedly as the parcel ascends, every 250–500 m when $w = 0.1$ m s$^{-1}$ (Fig. 18d) and at least ten times more frequently when $w = 0.01$ m s$^{-1}$ (Fig. 18f). At intermediate $w$, nucleation occurs roughly 50% less frequently for the slowest falling ice (Mitchell, Heymsfield) than for other ice properties. At the greatest $w$, nucleation does occur eventually if parcel ascent is continued for several kilometers (not shown). Thus, the frequency of nucleation events is impacted by the differing assumptions about ice properties and capacitance, and the spread in $N_i$ seen for $w = 1$ m s$^{-1}$ can be viewed as a frequency difference with a very long period. With sedimentation at $-55°$C, Fig. 17 shows that some size distributions happen to be in a period with small crystals present whereas others do not.

Although sedimentation only reduces parcel $m$ and $A_i$, maximum parcel $N_i$ may be increased over their values without sedimentation by more than a factor of two owing at least in part to faster aerosol freezing at colder temperatures. Nonetheless, in parcels subject to repeated nucleation events, sedimentation reduces time-averaged $N_i$ by nearly an order of magnitude and time-averaged $A_i$ by even more. $D_i$ and $\nu_i$ experience briefer discontinuities associated with nucleation events, $D_i$ dropping and $\nu_i$ increasing each time new crystals appear. The bullet and polycrystal models de-

rived here exhibit a lagged transition in $\nu_i$ after each nucleation event compared with the other ice properties, which can be attributed to evolution between $\rho_e$ varying not all with $D_{\max} < 100\ \mu$m (giving minimum $\nu_i$) to $\rho_e$ decreasing with $D_{\max} > 100\ \mu$m (increasing $\nu_i$ only when new crystals grow past $100\ \mu$m).

At the greatest $w$, sedimentation results in $D_i$ and $\nu$ similar in magnitude to that without sedimentation (e.g., Fig. 19f versus Fig. 19c), but the addition of sensitivity to fall speed increases the spread across $N_i$ and $A_i$. Thus, we conclude that with or without sedimentation, a chief effect of varying ice properties is on the size distribution of ice owing to differing $\rho_e$, leading to roughly factor of 2–3 differences in $D_i$ and $\nu_i$ in this parcel framework. We note that these parcel simulations with and without sedimentation generate results that span the range of $N_i$, $D_i$, $A_i$, and $\nu_i$ observed in situ during SPARTICUS, but we do not attempt any direct comparisons owing to the lack of realism of this simulation framework.

## 5.4    Optical properties

Extinction cross sections, scattering asymmetry parameters, and single-scattering albedos that are consistent with the derived crystal geometries are needed for interactive radiative calculations in cloud-resolving simulations and for calculation of diagnostic fluxes and radiances to be compared with measurements (e.g., van Diedenhoven et al., 2012). Infrared radiative transfer is dominated by emission, which is affected by particle size, but its sensitivity to crystal shape is minimal (e.g., Holz et al., 2016). However, particle shape does affect the relevant shortwave optical properties substantially. Detailed, accurate calculations of optical properties of nonspherical ice particles are generally computationally expensive. Existing databases and calculations of optical properties (e.g., Um and McFarquhar, 2007, 2011; Yang et al., 2013) assume crystal geometries based on sparse measurements and ad hoc assumptions that generally do not match the geometries derived here. As an alternative, approaches such as those of Fu (1996, 2007) and van Diedenhoven et al. (2014a) can be used to approximate the optical properties of complex crystals based on those of hexagonal prisms that serve as radiative proxies. Here we adopt the van Diedenhoven et al. (2014a) parameterization to approximate the optical properties of our derived crystal geometries. This parameterization provides the extinction cross section, asymmetry parameter, and single-scattering albedo at any shortwave wavelength for ice particles with any combination of crystal volume ($V = m/\rho_i$), and $A_p$, and $\alpha_e$ and roughness of crystal components. Ice refractive indices are taken from Warren and Brandt (2008).

The van Diedenhoven et al. (2014a) parameterization is based on geometric optics calculations. Accordingly, it assumes the extinction efficiency ($Q_e$) to be 2 for all particles and wavelengths. To partly correct this simplification for small particle sizes, here we apply anomalous diffraction theory to adjust $Q_e$ at wavelength $\lambda$ for particles with effective size parameter $P = 2\pi V(m_r - 1)/(\lambda A_p)$ less than $\pi/2$, where $m_r$ is the real part of the ice refractive index (Bryant and Latimer, 1969). We also apply the edge effect adjustment given by Nussenzveig and Wiscombe (1980). Both adjustments depend on $V$ and $A_p$.

The single-scattering albedo ($w_s$) is parameterized as a function of $V$, $A_p$, and $\alpha_e$ of the crystal components. All models use $\alpha_e$ of bullet arms for this calculation. In the case of the bullet and aggregate models, the arm length is taken to include the cap, and the width is taken as the cap base width where arms comprise only caps. In case of the polycrystal model, we use only bullet arm $\alpha_e$, neglecting the slight increase of $w_s$ owing to the thinness of the plates between arms. For the Bucky ball model, the $\alpha_e$ of the arms as given in Fig. 6c is limited to values of unity or greater to roughly account for the influence of the compact core where budding arms remain shorter than they are wide.

The asymmetry parameter ($g$) depends on particle $V$, $A_p$, and $\alpha_e$ values, as well as the crystal surface roughness, which may substantially lower $g$ (e.g., Macke et al., 1996; van Diedenhoven et al., 2014a). In the van Diedenhoven et al. (2014a) parameterization, the level of surface distortion is specified by a roughness parameter $\delta$ as defined by Macke et al. (1996). The Macke et al. (1996) ray-tracing code perturbs the normal of the crystal surface from its nominal orientation by an angle that, for each interaction with a ray, is varied randomly with uniform distribution between 0 and $\delta$ times $90°$. Similar commonly used parameterizations of particle roughness perturb the crystal surfaces using Weibull (Shcherbakov et al., 2006) or Gaussian (Baum et al., 2014) statistics rather than uniform distributions. However, Neshyba et al. (2013) and Geogdzhayev and van Diedenhoven (2016) demonstrated that the same roughness parameter value defined through a Weibull, Gaussian or uniform distribution represents very similar crystal microscale surfaces and yields largely equivalent scattering properties. Unfortunately, the roughness parameter cannot be constrained by the CPI data used here. Laboratory studies demonstrate that the microscopic structure of ice crystals is dependent on the environmental conditions in which they grow (Neshyba et al., 2013; Magee et al., 2014; Schnaiter et al., 2016). Since Baum et al. (2014) and van Diedenhoven et al. (2014b) show that a roughness parameter of 0.5 best fit observations, that is the default value we adopt here. For the Bucky ball model, we average core and arm $g$ values, weighted by their relative contributions to total $A_p$ (cf. Fu, 2007; van Diedenhoven et al., 2015). For the polycrystal model, the arm and plate $g$ values are averaged in the same way. Since the plate-like structures on the polycrystals shown in Fig. 10 appear relatively transparent, we assume smooth surfaces for the plates (i.e., $\delta = 0$).

Calculated $Q_e$, $g$, and $w_e$ are shown in Fig. 22 as a function of crystal $D_{\max}$. Also shown are the optical properties of the six-branch bullet rosette model calculated by Yang et al.

(2013). The geometry of the bullet rosettes assumed by Yang et al. (2013) is taken from Mitchell and Arnott (1994) and is similar to that of Mitchell et al. (1996) shown in Fig. 4. Yang et al. (2013) calculate the optical properties using a combination of improved geometric optics and other methods, which reveals resonances in the extinction efficiencies that are not seen in our results. However, such resonances largely cancel out when integrated over size distributions (Baum et al., 2014). The calculated $g$ values generally increase with size because of increasing $\alpha_e$ with size (cf. Fig 4c). At visible wavelengths, $g$ of the bullet, Bucky ball, and aggregate models, as well as the Yang et al. (2013) bullet rosettes, converge at about 0.81 at large sizes. Because of the addition of thin smooth plates to the polycrystal model, its $g$ is generally greater. Aside we note that assuming plates with $\delta = 0.5$ reduces 0.5-$\mu$m $g$ by only about 0.01 in the limit of large $D_{\max}$ (not shown), indicating that plate aspect ratio is the main cause of $g$ increase. At 2.1 $\mu$m, $g$ values increase owing to ice absorption (e.g., van Diedenhoven et al., 2014a). The 2.1-$\mu$m $g$ from Yang et al. (2013) is generally lower than our results because $w_s$ is generally greater. At a given $\lambda$, $w_s$ of an ice crystal is mostly determined by the particle effective diameter $D_{\mathrm{eff}} = 3V/(2A_p)$ (van Diedenhoven et al., 2014a). Figure 23 shows $D_{\mathrm{eff}}$ as a function of $D_{\max}$. For the Yang et al. (2013) bullet rosettes, $D_{\mathrm{eff}}(D_{\max})$ is generally substantially smaller than for our models, which is consistent with our generally greater $V$ and smaller $A_p$ compared to the Mitchell et al. (1996) bullet rosettes (see Fig. 4).

Figure 24 shows the shortwave optical properties integrated over the model size distributions at $-55°$C shown in Fig. 17. Extinction efficiencies generally increase slightly with $\lambda$ as $P$ decreases and are therefore generally greater for the cases with sedimentation owing to the smaller crystal sizes. At $\lambda \sim 2.8$ $\mu$m, a Christiansen band (Arnott et al., 1995) is present where a combination of strong absorption and refractive indices near or less than unity leads to a decrease in $Q_e$ (cf. Baum et al., 2014). The $w_s$ is generally greater for cases with sedimentation since these simulations lead to small $D_{\mathrm{eff}} \sim 20$ $\mu$m, whereas $D_{\mathrm{eff}}$ produced by simulations without sedimentation range from about 50 to 500 $\mu$m, primarily depending on $w$ (cf. Fig. 17). For the same reason, $g$ for cases with sedimentation is generally lower.

## 6 Discussion and conclusions

In preparation for large-eddy simulations with size-resolved microphysics for a case study of mid-latitude synoptic cirrus observed on 1–2 April 2010 during the SPARTICUS campaign (Muhlbauer et al., 2015), here we use CPI image analysis to develop ice crystal geometries that are physically continuous over the required crystal size range and suitable to calculate internally consistent physical and optical properties. The model to be used employs the Böhm (1999, 2004) approach to calculate fall speeds and pairwise collision rates

(based on crystal mass $m$, maximum projected area $A$, corresponding maximum dimension $D$, and bodily aspect ratio $\alpha$) and the van Diedenhoven et al. (2014a) approach to calculate radiative properties (based on crystal mass $m$, maximum projected area $A$, and crystal or polycrystal element aspect ratios $\alpha_e$). Assuming bullet rosettes as a typical geometry, we approximate $\alpha$ as unity (no preferred fall orientation), consistent with adoption of measured (randomly oriented) maximum dimension $D_{\max}$ and projected area $A_p$ for physical and optical properties. We then take an approach to estimating mass from CPI image data that begins with derivation of geometric crystal components suitable for calculation of optical properties, based on available ICR measurements. We also use derived $\alpha_e$ values in calculation of capacitance for vapor growth. This approach to ice crystal properties offers an advance over our past, ad hoc approach of using piecewise mass- and area-dimensional relations as a foundation, and then separately assigning aspect ratios based on sparse literature sources (e.g., Avramov et al., 2011; Fridlind et al., 2012; van Diedenhoven et al., 2012).

Our results using a typical bullet model of rosettes give $m(D_{\max})$ systematically larger than literature values used in similar past size-resolved microphysics simulations (Sölch and Kärcher, 2010), and $A_p(D_{\max})$ systematically smaller or similar. Taken together, these differences lead to $v_f$ greater by a factor of 1.5–2, and $w_s$ and $g$ respectively greater by about 0.2 and 0.05 in the limit of large $D_{\max}$ at near-infrared $\lambda$. A polycrystal model that estimates side plane growth on bullet rosettes increases $v_f$ by only about 15%, indicating that the effect of increased $m$ outweighs that of increased $A_p$ given the relatively ad hoc assumptions made here. In the polycrystal model, side plane growth also increases $g$ by about 0.05, primarily owing to plate aspect ratio.

In parcel simulations with and without sedimentation, differing ice properties lead to factors of 2–4 difference in crystal number concentration $N_i$, number-weighted mean diameter $D_i$, total projected area $A_i$, and size distribution relative dispersion $\nu_i$. When crystal effective density $\rho_e$ is smaller, $D_i$ is larger; when $\rho_e$ varies with size, $\nu_i$ is larger. When $v_f \gg w$, faster falling crystals are associated with more frequent nucleation events, by roughly 50% at $w = 0.1$ m s$^{-1}$.

Overall, it appears that the main differences between our models and past literature arise from differences in bullet rosette geometry (i.e., single-particle mass) or its representation (i.e., definition of $D_{\max}$). Where available, $A_p(D_{\max})$ and arm $\alpha_e(D_{\max})$ appear more similar, by contrast. Based on ad hoc assumptions made here, the chief potential impact of side plane growth could be an increase in $g$ by $\sim 0.05$ in the mid-visible. More detailed observational analysis would be needed to confirm side plane properties assumed here. However, differences between our polycrystal and bullet properties are surprisingly substantially less than the differences between our bullet properties and those in past literature, which may prioritize better establishing the baseline bullet rosette

model over working out details of irregular crystal properties.

Evolution of newly nucleated ice crystals may proceed from amorphous shapes to more defined habits (e.g., Baker and Lawson, 2006b; Schnaiter et al., 2016) in a manner that may depend in part on nucleation mode (e.g., Bacon et al., 2003; Schnaiter et al., 2016), but observations considered here are inadequate to derive a robust geometric model for $D_{max}$ smaller than roughly 100 $\mu$m, as in other recent work (e.g., Erfani and Mitchell, 2016). However, we find that growth from a budding Bucky ball shape versus an idealized bullet rosette shape could lead to non-negligible differences in normalized capacitance of nearly 0.1 (cf. Fig. 15). If such geometry is important to predicted PSD evolution, deriving a statistically decreasing number of arms with increasing size could be needed to simultaneously represent the evolution of crystal $m$, $A_p$, and $\alpha_e$. Or more accurate geometries could be established (e.g., Nousiainen et al., 2011; Schnaiter et al., 2016) and relevant physical and optical properties made appropriately consistent and continuous for modeling purposes. Evident diversity of both small and large crystal properties at a given $D_{max}$, even when most rigorously defined, could also be relevant.

It may be the case that uncertainties in ice crystal $m$ and its relationship to morphological properties, which together determine factors such as $v_f$ and radiative properties, are not sufficiently considered in current literature. Single-crystal mass measurements that were made laboriously in studies decades ago (e.g., Kajikawa, 1972; Mitchell et al., 1990) have not been replaced by improved measurements or substantially augmented since that time. In the case of bullet rosettes, for instance, we are aware of only one unpublished data set comprising 45 crystals, we are aware of no such measurements made at cirrus elevations, and it appears that those ground-level measurements may be biased to fewer branches, as discussed above. From analysis of the SPARTICUS data here, we can see that such a bias in branch number could likely be correlated with a bias in $\alpha_e$ and $m$. The degree to which a single habit-independent $m - A_p$ power law applied to 2DS PSDs leads to accurate calculation of $m(D_{max})$ for both anvil and synoptic cirrus crystal conditions may also warrant additional investigation (cf. Fig. 4e). As discussed by Baker and Lawson (2006a), for instance, particles are not entirely randomly oriented in the petri dish measurement approach used in the SCPP data set; to the extent that non-random orientation favors a higher ratio of $A_p/m$ on a petri dish, the derived $m(A_p)$ could be biased correspondingly low when applied to randomly oriented crystal images.

With respect to classification of morphological properties, it also appears to be the case that classification algorithms may give substantially differing results. For instance, whereas here roughly 80% of crystals with $D_{max}$ greater than 100 $\mu$m are unclassified (irregular), the algorithm reported by Lindqvist et al. (2012) classifies more than 50% of crystals as rosettes in a similar mid-latitude cloud. The fact that their study places fewer than 20% of crystals in an irregular class across tropical, Arctic, and mid-latitude conditions suggests that it is fundamentally different from the algorithm applied here. The fact that unclassified crystals here differ relatively little in derived properties from bullet rosettes (with the possible exception of $g$, given some relatively ad hoc assumptions) suggests that algorithms may currently differ in the allowable degree of deviation from a pristine state. It may be useful to establish comparable statistics from differing algorithms to allow comparison of circumference or other non-habit-dependent measures.

Overall, the results obtained here motivate the use of our derived ice properties in comparison with more widely used values in 3D simulations of the April 1–2 SPARTICUS conditions, which can in turn be compared with in situ ice size distribution observations.

## Appendix A: Ice crystal models

A fundamental geometric element of all ice models considered below is the regular hexagonal column with length $L$ and width $W$, defined here as twice hexagon side length. In all cases the true mean branch width $W$ is taken as the mean of the measured widths of all branches divided by a factor of $(1 + \sqrt{3}/2)/2$ to correct for random orientation. Thus, true mean branch width $W$ is about 7% wider than measured mean branch width.

Since branch length measurements extend from crystal center to the outermost edge of projected randomly oriented branches, the true mean total branch length (including cap or core contributions, depending on the model) is taken as the mean of the measured lengths less one-half of the mean of the measured widths times $\pi/4$ (the contribution of randomly oriented projected base to measured length), all multiplied by a factor of $4/\pi$ to account for branch foreshortening by random orientation. Thus, true mean branch length is about 30% longer than measured mean branch length corrected for the contribution of column base projection.

All non-aggregate models (bullet, Bucky ball, and polycrystal) assume six branches, consistent with mean and median number found over all bullet rosettes measurable with the ICR software.

Derived ice properties are supplied as the Supplement.

### A1    Bullet model

The bullet model assumes that each hexagonal column has a single cap, and that the six caps meet at a point in the center of the crystal. If the cap is a hexagonal pyramid with a fixed angle $\beta$ between pyramid edges and the line defining pyramid height, then cap length $L_c$ scales with column width according to

$$L_c = \frac{W}{2\tan\beta}. \tag{A1}$$

Thus, wider branches have longer caps. Here we assume fixed $\beta$, and assign a value of $22°$; generally wider angles have been assumed in previous work, as discussed further at the end of this section.

For the bullet model, total true branch length includes both hexagonal column length $L$ and hexagonal pyramid cap length ($L_c$). A least squares linear fit of total mean branch length $L + L_c$ to measured maximum dimension (uncorrected for random orientation throughout, see Section 3) gives a line nearly through the origin. Adopting only the slope (cf. Fig. 4a) gives

$$L + L_c = 0.691 D_{\max}. \tag{A2}$$

A least squares fit of mean branch width $W$ to $D_{\max}$ gives (cf. Fig. 4b, $\mu$m units)

$$W = 0.139 D_{\max} + 40.6. \tag{A3}$$

In the limit of zero $D_{\max}$, Equation A3 would give a non-zero branch width. As a simple physical solution, we assume that branch width is equal to cap base width wherever predicted cap length per Equations A1 and A3 is greater than total branch length per Equation A2, designated as $D_{max,L0}$ (where branch length $L$ is zero). Thus, for the bullet model, crystals with mean branch width less than $D_{max,L0} \sim 50$ $\mu$m comprise hexagonal pyramids without developed hexagonal columns extending from them; we note that no ICR measurements were possible at such small sizes.

With crystal geometry now defined using the bullet model, it is straightforward to calculate the aspect ratio $W/(L + L_c)$, surface area $A_s$, and mass $m$ for each measured crystal (symbols in Fig. 4c–e). The crystal model derived and used in simulations is that for a corresponding typical crystal with number of branches fixed to six, equal to both mean and median of measured branch numbers (solid lines in Fig. 4c–e), with mass therefore defined as

$$m = 6\rho_i \frac{\sqrt{3}}{8} W^2 (3L + L_c), \tag{A4}$$

where $\rho_i$ is the bulk density of ice, taken here as $0.917$ g cm$^3$, and surface area defined as

$$A_s = 6 \left( 3LW + \frac{3\sqrt{3}}{8} W^2 + \frac{3}{4} W \sqrt{\frac{3}{4} W^2 + 4L_c^2} \right). \tag{A5}$$

However, the randomly oriented projected area $A_p$ is not analytically defined. A least squares fit of measured crystal projected areas to bullet model crystal surface areas results in a line nearly through the origin (cf. Fig. 4d), and this is used with Equation A5 to define model projected area

$$A_p = \lambda_b A_s, \tag{A6}$$

where $\lambda_b = 0.107$. The linear relationship and slope $< 0.25$ are consistent with theory for convex particles (Vouk, 1948), as discussed above.

If the cap angle $\beta$ is increased, effective density and projected area ratio increase for ice crystals with $D_{\max}$ smaller than about 90 $\mu$m. In the limit of small $D_{\max}$, a $\beta$ of $22°$ is selected to give effective density and projected area ratio no larger than that calculated for any measured rosettes (cf. Fig. 5a, b). Calculated fall speeds are not strongly sensitive to changes in $\beta$ because effective density and projected area increase or decrease together. Regarding choice of $\beta$, Iaquinta et al. (1995) have noted that a bullet rosette with a six-faced pyramidal end and a 56-° angle between opposing faces has been assumed in past work but cannot fit to form a multibranched bullet rosette, leading to their adoption of a trilateral pyramidal end as "only an idealized form of the sharp end of natural ice crystals"; we select $\beta$ values here in the same spirit.

## A2 Bucky ball model

The Bucky ball model assumes that each hexagonal column grows initially from a Bucky ball face. The core is approximated as a sphere with diameter $D_c$ of 10 $\mu$m, and budding columns are assigned an initial width $W_{min}$ of 4 $\mu$m. In order to insure a branch length of zero when $D_{\max}$ is equal to that of a sphere with core diameter $D_c$, a slope is fit to $L$ as a function of $D_{\max} - D_c$ (Fig. 6a), giving

$$L = 0.684(D_{\max} - D_c). \tag{A7}$$

Similarly, in order to insure a branch width of $W_{min}$ when the maximum dimension is equal to the core diameter $D_c$, a slope is fit to $W - W_{min}$ as a function of $D_{\max} - D_c$ (Fig. 6b), giving

$$W - W_{min} = 0.216(D_{\max} - D_c). \tag{A8}$$

With crystal geometry now defined using the Bucky ball model, it is straightforward to calculate the branch aspect ratio ($L/W$) and mass for each measured crystal. For the canonical crystal with six arms, where $D_{\max} < D_c$, then values are those of a sphere with diameter $D_{\max}$ and density $\rho_i$. Otherwise, using $W$ and $L$ from Equations A7 and A8,

$$m = \rho_i \left( \frac{\pi}{6} D_c^3 + 6 \frac{3\sqrt{3}}{8} W^2 L \right). \tag{A9}$$

Rigorous calculation of $A_s$ for the model crystal with typical six branches is less straightforward. Here we take the ad hoc approach of first estimating the surface area of measured crystals as the total of branches with one end each, neglecting the inner end faces (Equation A5 without the third term that represents cap surface area). A fit of $A_p$ measured to $A_{s,est}$ so estimated gives a slope 0.0921 (Fig. 6d). The model crystal with six arms is then assigned $A_s(D_{\max})$ as a weighted average of estimated $A_s$ and that of a sphere with diameter $D_{\max}$,

$$A_s = f_s 4\pi D_{\max}^2 + f_b A_{s,est}, \tag{A10}$$

where $m_r$ is the ratio of $m$ to that of a sphere with diameter $D_{\max}$, $m_{r,max}$ is the value in the limit of large $D_{\max}$ (roughly 0.24),

$$f_s = 1 - \frac{1-m_r}{1-m_{r,max}}, \tag{A11}$$

and

$$f_b = 1 - f_s. \tag{A12}$$

Relative to the bullet model, the Bucky ball model exhibits a stronger increase of $W$ with increasing $D_{\max}$ (cf. Fig. 4b and 6b) and a nearly constant aspect ratio at $D_{\max}$ greater than 100 $\mu$m (cf. Fig. 4c and 6c). Smooth variation of radiative properties and capacitance in the limit of small $D_{\max}$ is achieved with

$$\alpha_e = max(1, L/W) \tag{A13}$$

and (Fig. 15)

$$\frac{C}{D_{\max}} = min\left(0.5, 0.4\left(\frac{L}{W}\right)^{0.25}\right). \tag{A14}$$

### A3 Aggregate model

The 'aggregate model' is an extension of the bullet model. A least squares fit of mean branch length $L+L_c$ to $D_{\max}$ gives a line again nearly through the origin. Adopting only the slope as in the bullet model (cf. Fig. 9a) gives

$$L + L_c = 0.461 D_{\max}. \tag{A15}$$

The mean and median measured number of arms is twelve, consistent with aggregates primarily of two typical single rosettes. The roughly 30% reduction in slope compared with single rosettes can be attributed to the overlap of aggregate arms, compounded by random orientation when two crystals create a linearly aligned pair that will be rarely normal to the viewing angle. A least squares fit of mean branch width $W$ to $D_{\max}$ gives (cf. Fig. 9b, $\mu$m units)

$$W = 0.0886 D_{\max} + 44.9. \tag{A16}$$

To handle unphysical branch widths in the limit of zero $D_{\max}$, we again assume that branch width is equal to cap base width wherever predicted cap length per Equations A1 and A16 would be greater than total branch length per Equation A15.

Using this model for aggregates, mass and projected area are simply twice that of bullet rosettes,

$$m = 12\rho_i \frac{\sqrt{3}}{8} W^2 (3L + L_c), \tag{A17}$$

where $\rho_i$ is the bulk density of ice, taken here as 0.917 g cm$^3$, and surface area defined as

$$A_s = 12\left(3LW + \frac{3\sqrt{3}}{8}W^2 + \frac{3}{4}W\sqrt{\frac{3}{4}W^2 + 4L_c^2}\right). \tag{A18}$$

Using Equation A18 with A1, A15 and A16 to calculate $A_s$, measured $A_p$ is found to be 10% of calculated $A_s$ (cf. Fig. 9d), roughly 1% lower than found for single rosettes using the bullet model, consistent with branch entanglement that reduces $A_p$ but not $A_s$ relative to a pair of single rosettes.

### A4 Polycrystal model

The polycrystal model is derived for unclassified crystals using plate growth on the bullet model as a basis. When the unclassified crystals are initially assumed to follow the bullet model, and measured $A_p$ is regressed against calculated bullet surface area ($A_{s,b}$) following Equation A5, assuming six arms per crystals, the slope is greater than found for the bullet model, consistent with systematically greater projected area than rosettes demonstrated in Fig. 12. We adopt the ad hoc assumption that additional projected area can be attributed to side plane growth, represented here for simplicity as growth of hexagonal plates. Continuing with the six-arm bullet model as a basis, we further make the ad hoc assumption that a single plate is grown on each arm with sufficient total plate surface area ($A_{s,p}$) to restore $A_p/(A_{s,b} + A_{s,p})$ to a value near $\lambda_b$ in the limit of large $D_{\max}$.

Based on trial and error, taking the foregoing assumptions as a recipe, the following prescription was found to match effective density from Cotton et al. (2012) in the limit of small $D_{\max}$ (Fig. 13e) and median measured $A_p(D_{\max})$ for unclassified crystals at all sizes (Fig. 13f) to the extent possible without exceeding the effective diameter of equivalent-sized spheres. First, to increase effective density relative to the bullet model where crystals are entirely truncated caps ($D_{\max} < D_{max,L0}$), $\beta$ is increased to 25°. If $A_{s,b}$ is then calculated following Equation A5 for measured crystals, a slope $\lambda_i = 0.147$ is found (Fig. 13a), larger than $\lambda_b = 0.107$ found for rosettes using the bullet model for measured rosettes (Equation A6). Next $A_p/A_s$ is matched to allow zero plate contribution to surface area where $D_{\max} < D_{max,L0}$ (noting that increased $\beta$ slightly reduces $D_{max,L0}$ relative to that for the bullet model) and maximum contribution to surface area within an ad hoc scale length of $2D_{max,L0}$ using

$$\Delta D = D_{\max} - D_{max,L0} \tag{A19}$$

and, taking $\lambda_p = 0.1$ as the ratio of $A_p$ to $A_s$ for polycrystals (reduced by an ad hoc amount from that for bullets on the basis that $A_s$ has increased relatively more than $A_p$, but not so much that effective density exceeds that of equivalent-sized spheres),

$$\lambda = \lambda_p + (\lambda_i - \lambda_p)\left(1 - exp\left(-\frac{\Delta D}{2D_{max,L0}}\right)\right). \tag{A20}$$

For the model crystal with six arms, the plate contribution to surface area is then

$$A_{s,p} = 6\left(\frac{\lambda}{\lambda_p} - 1\right) A_{s,b}. \tag{A21}$$

If plate surface area is approximated as twice the face areas (neglecting edge contributions), then per-plate diameter ($D_p$), defined for $D_{\max} > D_{max,L0}$, can be calculated from

$$6D_p = \sqrt{\frac{2A_{s,p}}{9\sqrt{3}}}. \tag{A22}$$

Plate thickness $L_p$, which is neglected in the addition of plate surface area to $A_s$ but is included in calculation of plate contribution to crystal mass, is taken as (Pruppacher and Klett, 1997, their Table 2.2a, cm units):

$$L_p = \min\left(0.1D_p, 0.0141D_p^{0.474}\right). \tag{A23}$$

In the limit of zero plate size, $L_p$ is not permitted to exceed $0.1D_p$. Thus, for radiative calculations, the maximum plate aspect ratio $\alpha_{e,p} = L_p/D_p = 0.1$ and the bullet arm aspect ratio remains as $\alpha_{e,b} = (L + L_c)/W$. Normalized capacitance is calculated as for a bullet rosette with $L(\beta)$ and $W(\beta)$, neglecting the presence of plates, for lack of another obvious strategy.

Plate contribution to polycrystal mass can be calculated as

$$m_p = \frac{A_{s,p}}{2}L_p. \tag{A24}$$

Total mass is then $m_p$ plus bullet mass following Equation A4 with $\beta = 25°$, and total projected area $A_p = \lambda(A_{s,p} + A_{s,b})$.

*Acknowledgements.* This work was supported by the NASA Radiation Sciences Program and the Office of Science (BER), U.S. Department of Energy under agreements DE-SC0006988, DE-SC0008500, and DE-SC0014065. This research used resources of the National Energy Research Scientific Computing Center, a DOE Office of Science User Facility supported by the Office of Science of the U.S. Department of Energy under Contract No. DE-AC02-05CH11231. Resources supporting this work were also provided by the NASA High-End Computing (HEC) Program through the NASA Advanced Supercomputing (NAS) Division at Ames Research Center. We thank the SPARTICUS science team for collecting and archiving all data sets referenced.

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

**Table 1.** Mass- and area-dimensional power law coefficients for $m = aD_{max}^b$ and $A = cD_{max}^d$ in cgs units.

| Habit | $D_{max}$ | $a$ | $b$ | $c$ | $d$ | Source[†] |
|---|---|---|---|---|---|---|
| Small bullet rosettes | <0.01 | 0.1 | 2.997 | 0.629535[†] | 2.0[†] | Mitchell (1994) |
| Large bullet rosettes | 0.02–1 | 0.00308 | 2.26 | 0.08687 | 1.568 | Mitchell (1996) |
| Large bullet rosettes | 0.02–2 | 0.0139 | 2.54 | 0.2148[‡] | 1.7956[‡] | Heymsfield et al. (2002) |
| Bullet rosette aggregates | 0.04-2 | 0.00183 | 2.04 | 0.0803 | 1.45 | Heymsfield et al. (2002) |

[†] Calculated as described in text.
[‡] As cited in Sölch and Kärcher (2010), see text.

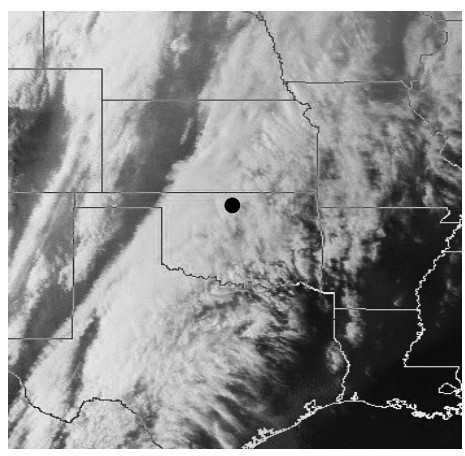

**Figure 1.** GOES composite image at 23:39 UTC on 1 April 2010. Black circle indicates the Southern Great Plains long-term measurement site in Oklahoma.

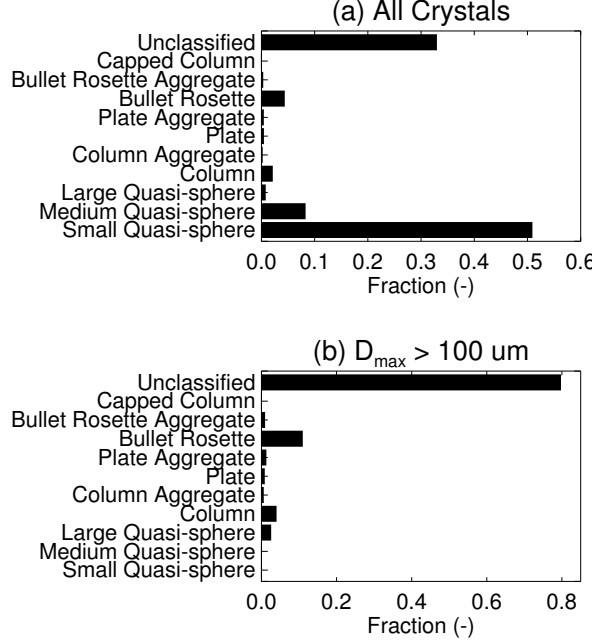

**Figure 2.** Fraction of ice by habit class for all crystals imaged (a) and for those with $D_{max}$ greater than 100 $\mu$m (b).

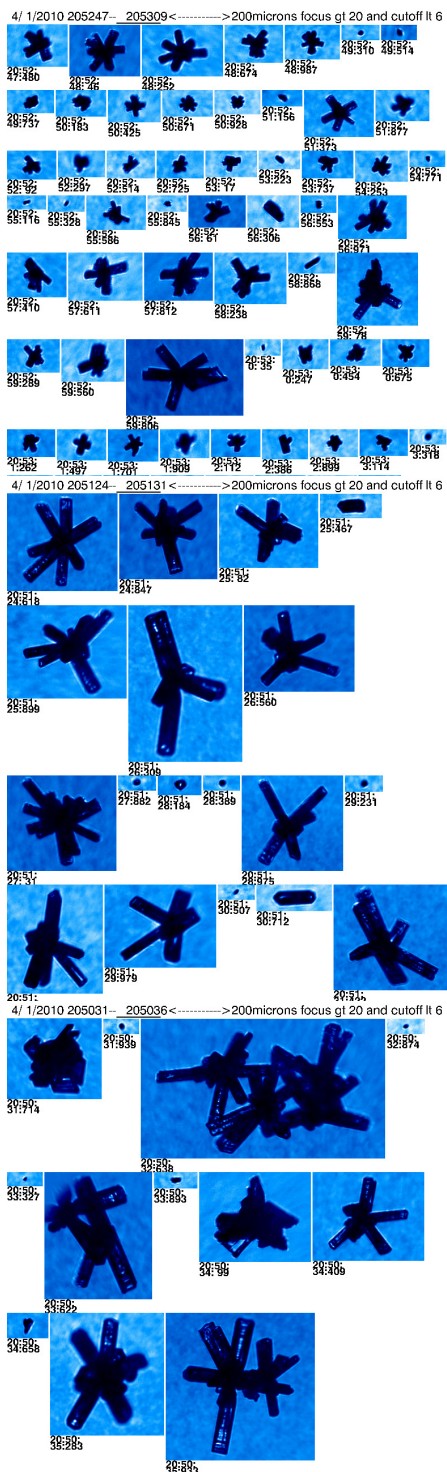

**Figure 3.** Bullet rosettes imaged on 1 April with $D_{\mathrm{max}}$ commonly smaller than 200 $\mu$m (top), larger than 200 $\mu$m (middle), and aggregated (bottom).

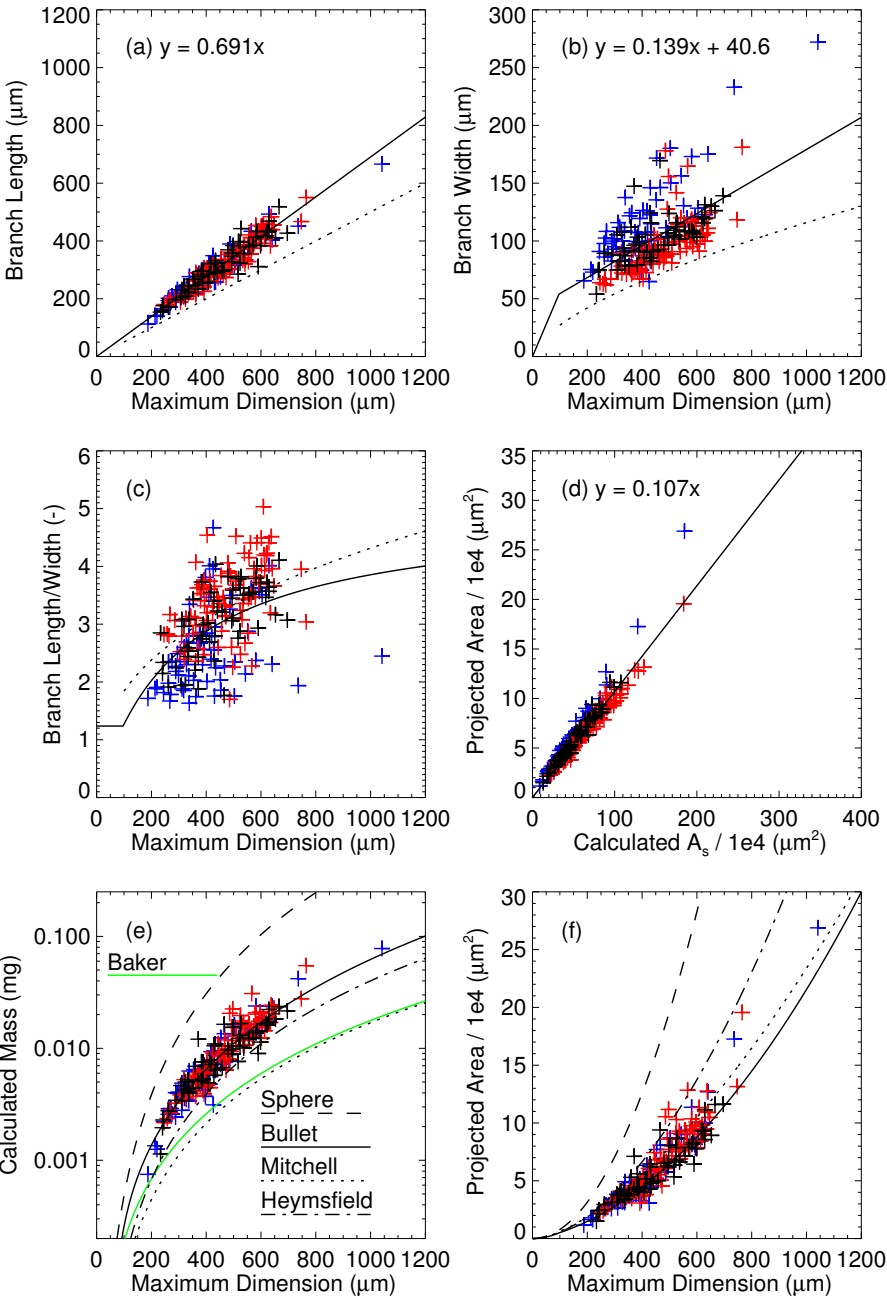

**Figure 4.** Bullet model: measured and calculated properties of imaged bullet rosettes with six arms (black symbols), fewer than six arms (blue symbols), and more than six arms (red symbols). Line types indicate derived ice properties as follows (see legend in panel e): a sphere, a six-arm bullet rosette per the bullet model (see Section 4.1 and Appendix A1), five-arm rosettes from Mitchell et al. (1996), and cirrus crystals from Heymsfield et al. (2002). Also shown is the habit-independent $m - A_p$ relation derived by Baker and Lawson (2006a).

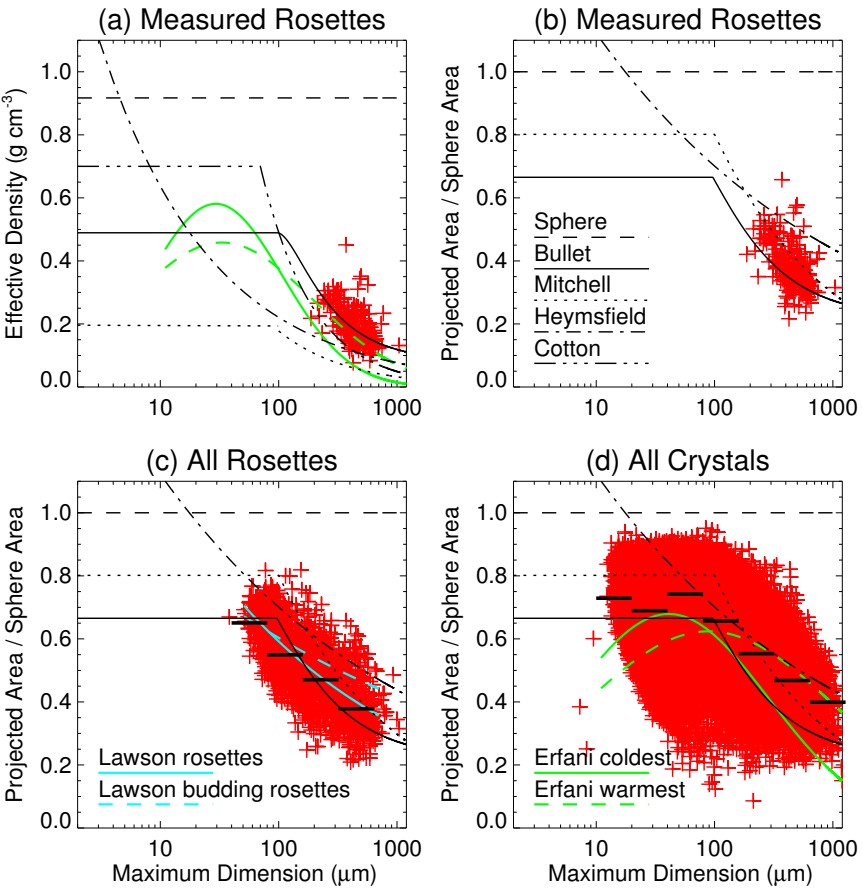

**Figure 5.** Bullet model: measured and calculated properties of imaged ice crystals (red symbols) emphasizing the transition to the smallest sizes. Effective density and projected area for bullet rosettes with ICR measurements (a, b), and projected area for all bullet rosettes identified (c, including those not measurable with the ICR software), and for all crystals imaged during the April 1–2 flights (d). Within $D_{max}$ doubling bins, the median of measurements is shown where a bin contains more than 100 measurements (thick solid line segments, c and d only). Other line types indicate derived ice properties as follows (see legend in panel b): a sphere, a six-arm bullet rosette per the bullet model (see Section 4.1 and Appendix A1), five-arm rosettes from Mitchell et al. (1996), and cirrus crystals from Heymsfield et al. (2002) and Cotton et al. (2012). Also shown (see legends in c and d): fits to measured areas of bullet rosettes and budding bullet rosettes from Lawson et al. (2006a), and polynomial fits from Erfani and Mitchell (2016) for synoptic cirrus crystals at −55 to −65°C (coldest range fitted) and −20 to −40°C (warmest, see text).

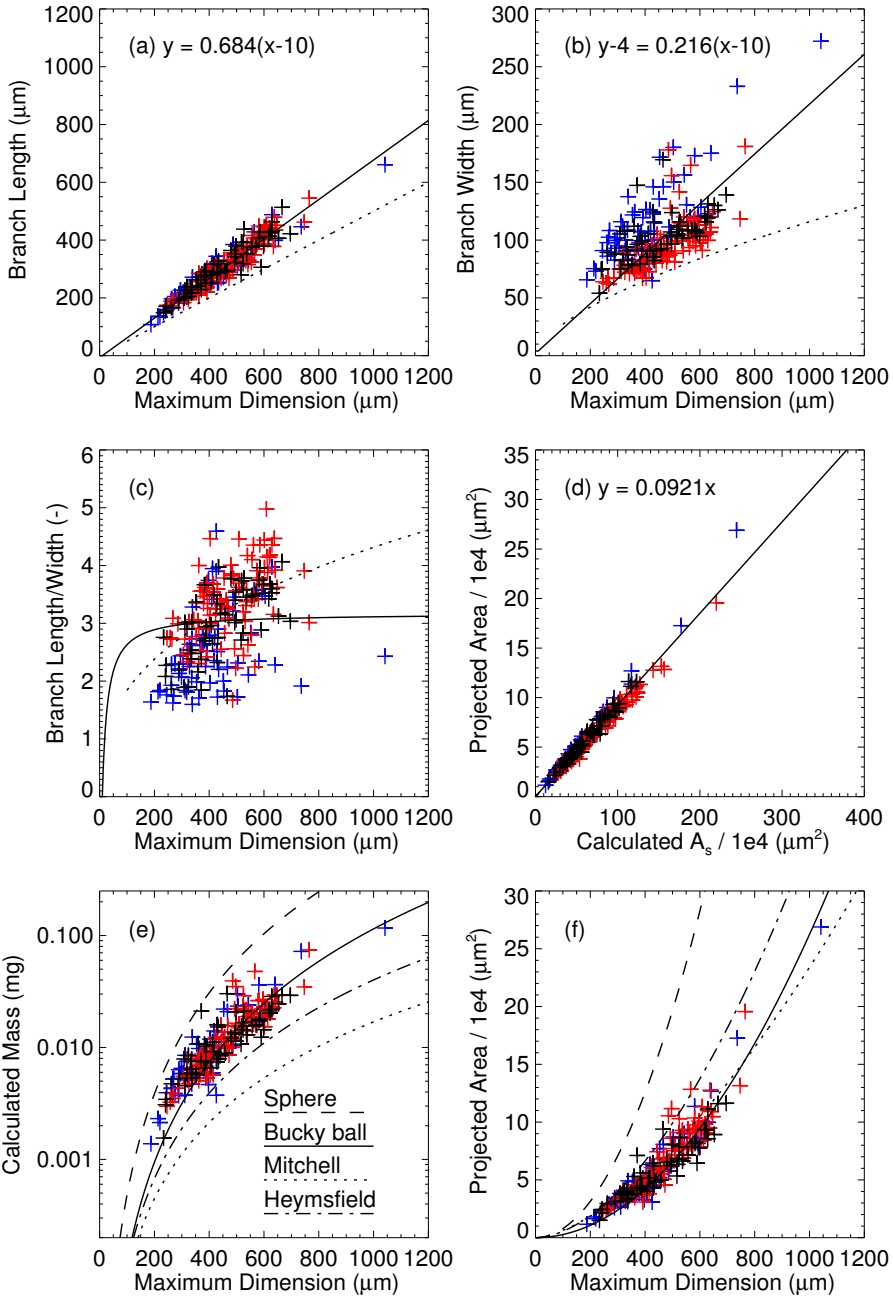

**Figure 6.** As in Fig. 4 except for Bucky ball model (see Section 4.2 and Appendix A2).

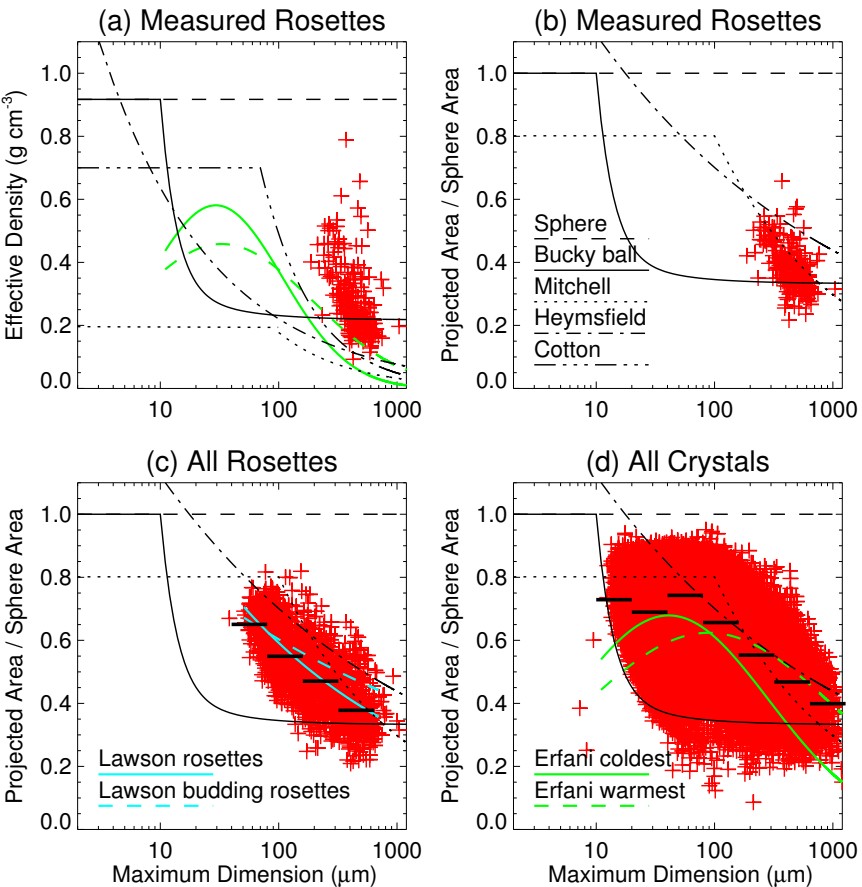

**Figure 7.** As in Fig. 5 except for Bucky ball model.

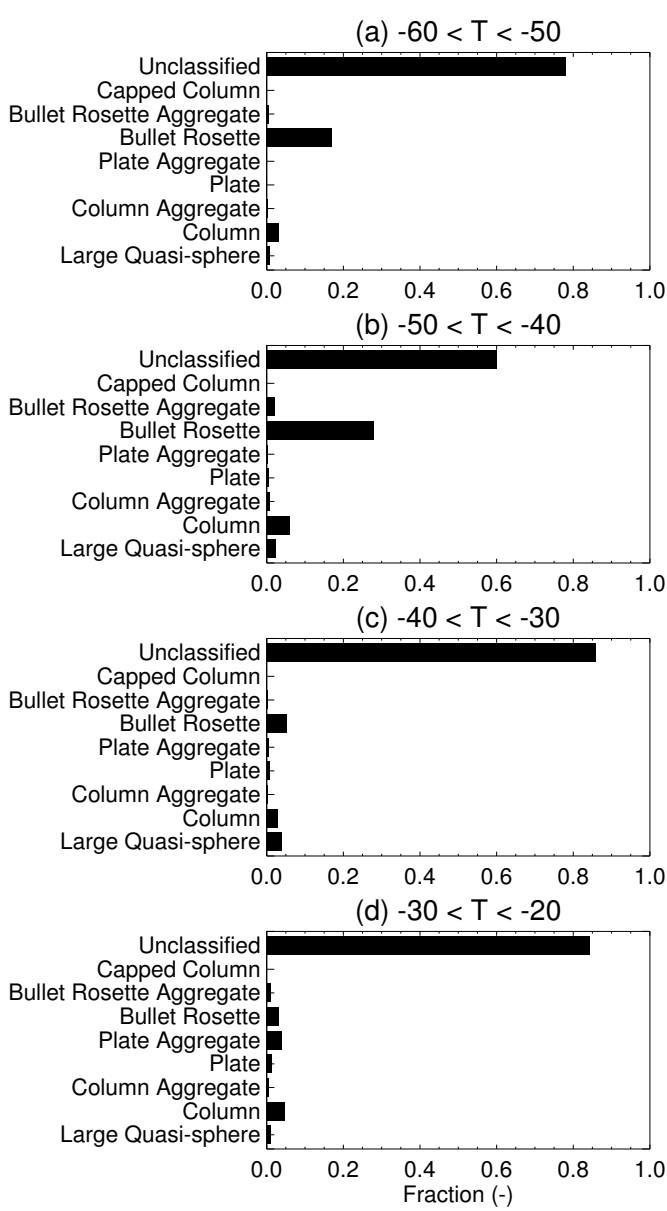

**Figure 8.** Fraction of imaged ice crystals with $D_{\mathrm{max}}$ greater than 100 $\mu$m in four temperature ranges in degrees Celsius.

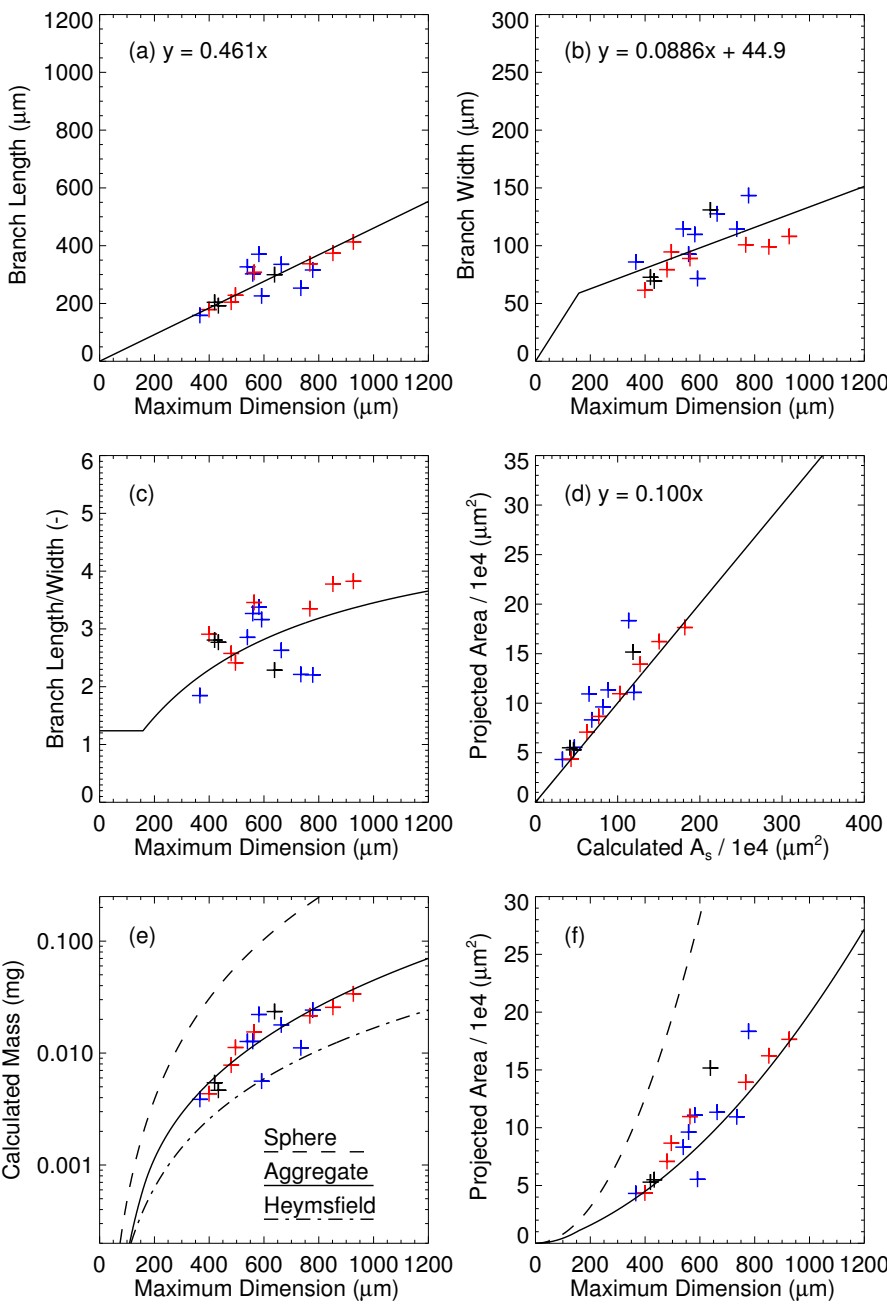

**Figure 9.** Aggregate model: measured and calculated properties of aggregates of bullet rosettes with twelve arms (black symbols), fewer than twelve arms (blue symbols), and more than twelve arms (red symbols). Line types indicate derived ice properties as follows (see legend in panel e): a sphere, a twelve-arm bullet rosette aggregate (see Section 4.3 and Appendix A3), and aggregates of bullet rosettes from Heymsfield et al. (2002).

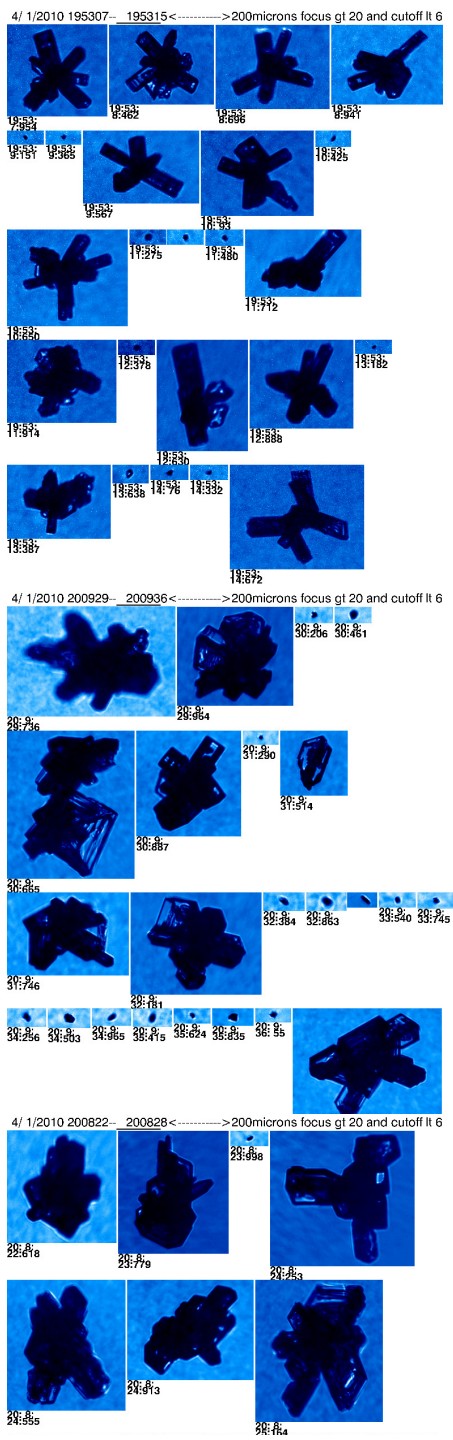

**Figure 10.** Bullet rosettes and unclassified crystals with radiating growth imaged on 1 April.

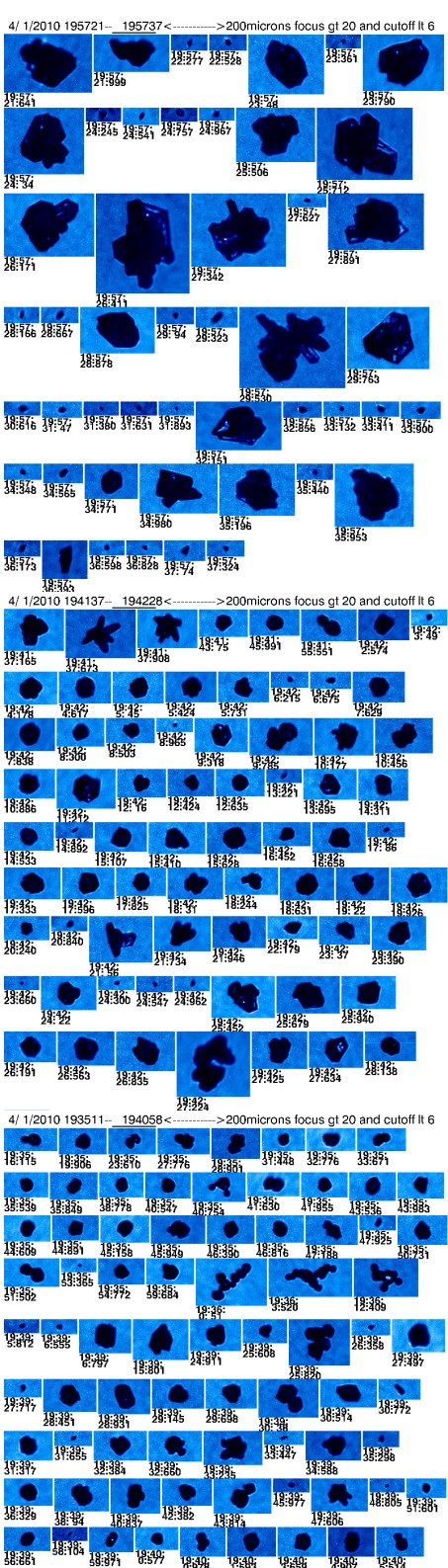

**Figure 11.** Unclassified ice crystals with sublimated edges imaged on 1 April.

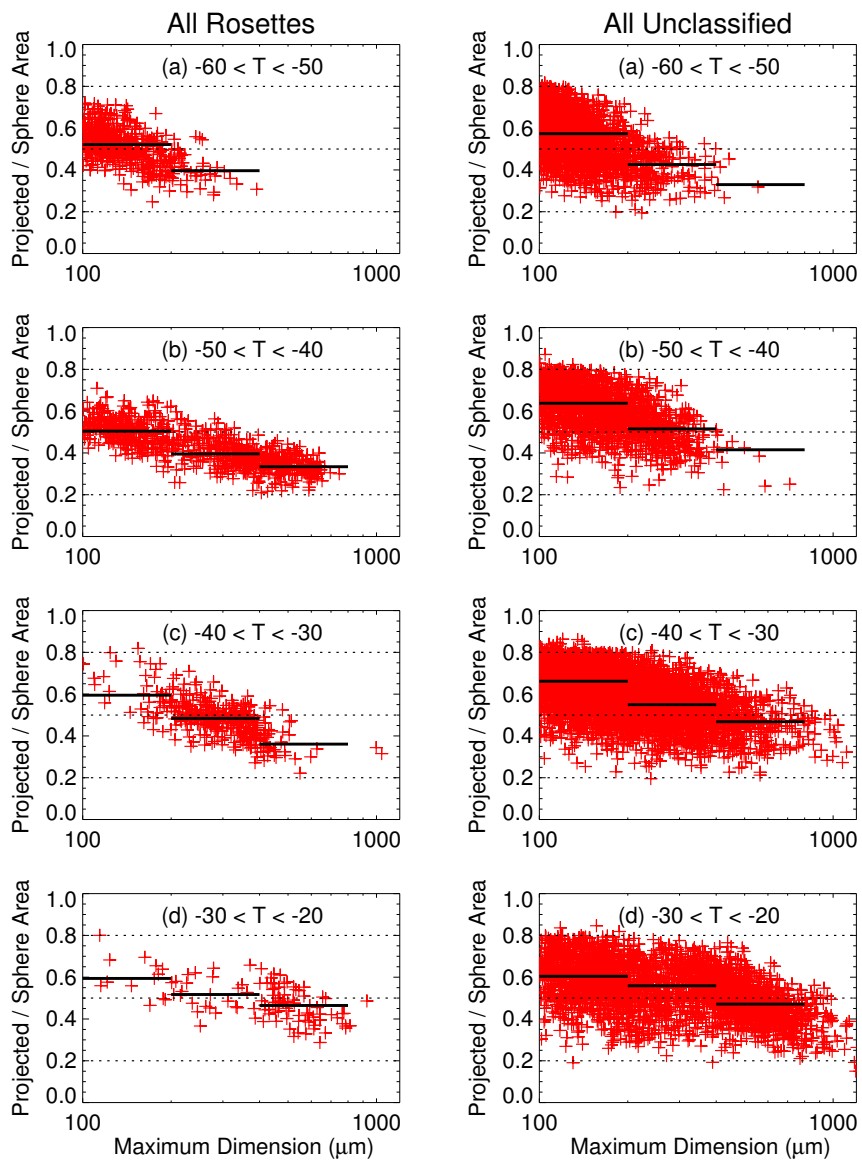

**Figure 12.** Ratio of measured $A_p$ to area of a sphere with diameter $D_{\max}$ in four temperature ranges for all bullet rosettes (left column) and all unclassified crystals (right column). Overplotted solid line segments indicate median value over $D_{\max}$-doubling bins.

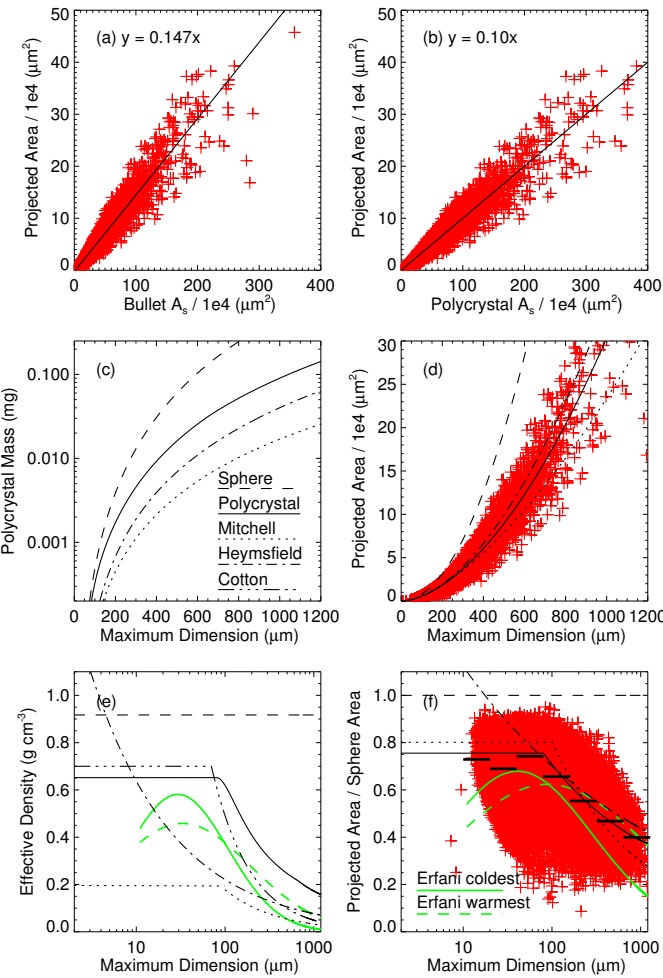

**Figure 13.** Polycrystal model: measured and calculated properties of unclassified ice crystals (red symbols). Line types indicate derived ice properties as follows (see legends in panels c and f): a sphere, a polycrystal based on a six-arm bullet rosette (see Section 4.4 and Appendix A4), five-arm rosettes from Mitchell et al. (1996), and cirrus crystals from Heymsfield et al. (2002), Cotton et al. (2012), and Erfani and Mitchell (2016).

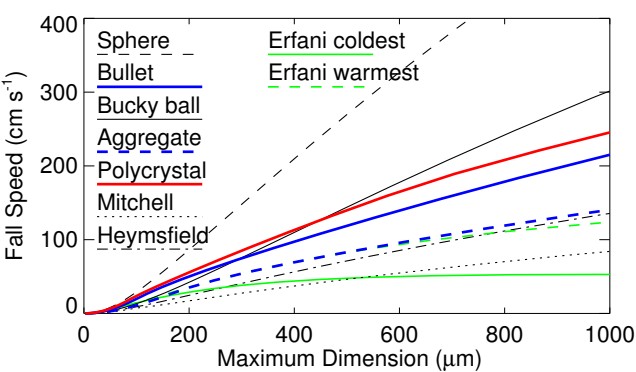

**Figure 14.** Ice crystal fall speeds at 350 mb and 233 K for derived ice properties as follows (see legend): a sphere, six-arm rosettes following the bullet and Bucky ball models, twelve-arm aggregates following the bullet model, the polycrystal model, five-arm rosettes from Mitchell et al. (1996), and cirrus crystals from Heymsfield et al. (2002) and from Erfani and Mitchell (2016) assuming ice crystal properties at $-55$ to $-65°$C (coldest range fitted) and $-20$ to $-40°$C (warmest, see text).

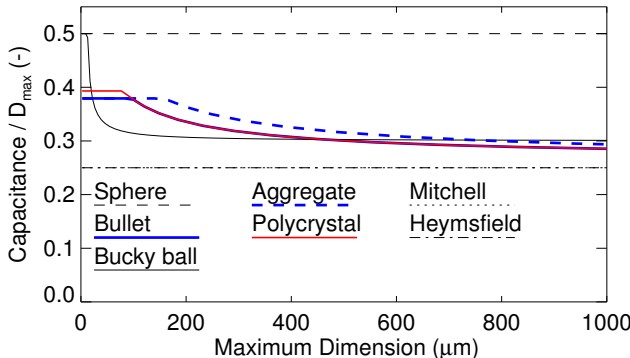

**Figure 15.** Ice crystal capacitance normalized by maximum dimension at 350 mb and 233 K for derived ice properties as in Fig. 14. In the absence of specified $\alpha_e$ for some or all crystal sizes, a constant value is taken for Mitchell et al. (1996) and Heymsfield et al. (2002) ice properties (see text).

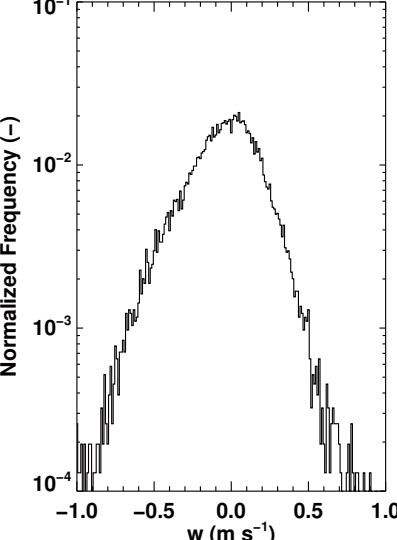

**Figure 16.** Vertical wind speeds retrieved from 19:06 UTC on 1 April to 2:23 UTC on 2 April at elevations of 6.1–12.0 km, from a sample size of 123,469 retrievals obtained in 47 layers at 10-s resolution.

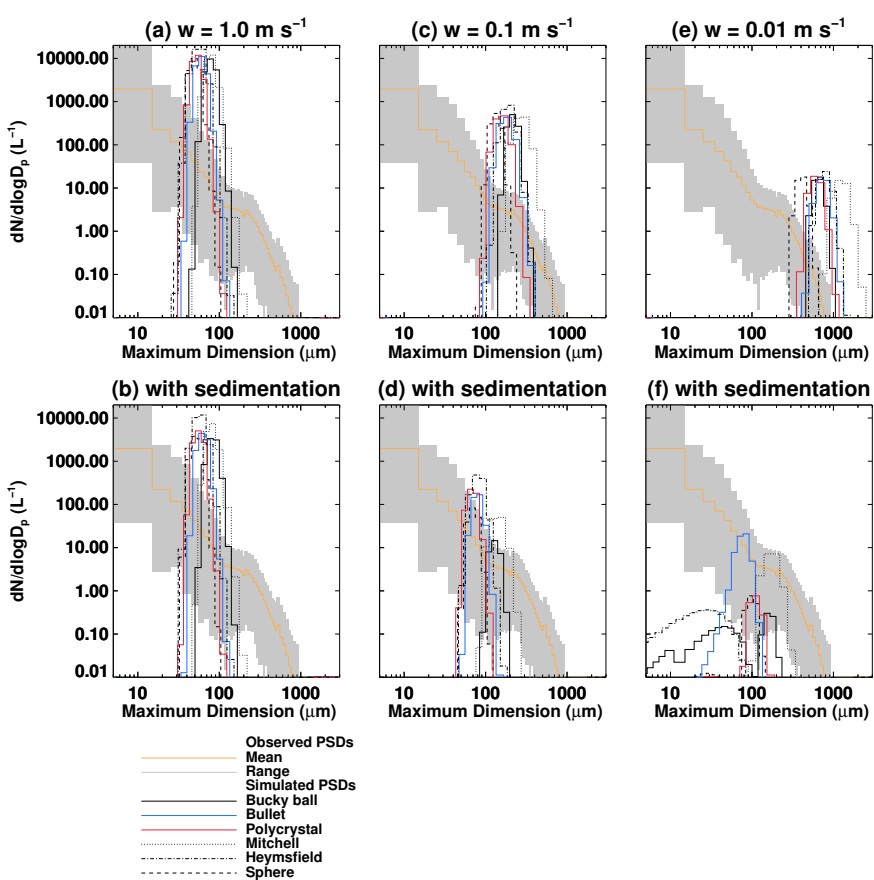

**Figure 17.** Ice particle size distributions (PSDs) simulated at $-55^\circ$C with different ice properties (see legend) and different updraft speeds without sedimentation (top row) and with sedimentation (bottom row). For context are shown also the mean and range of all PSDs observed over the 1–2 April flights using an in-cloud ice water content threshold of 0.001 g m$^{-3}$ following Jackson et al. (2015).

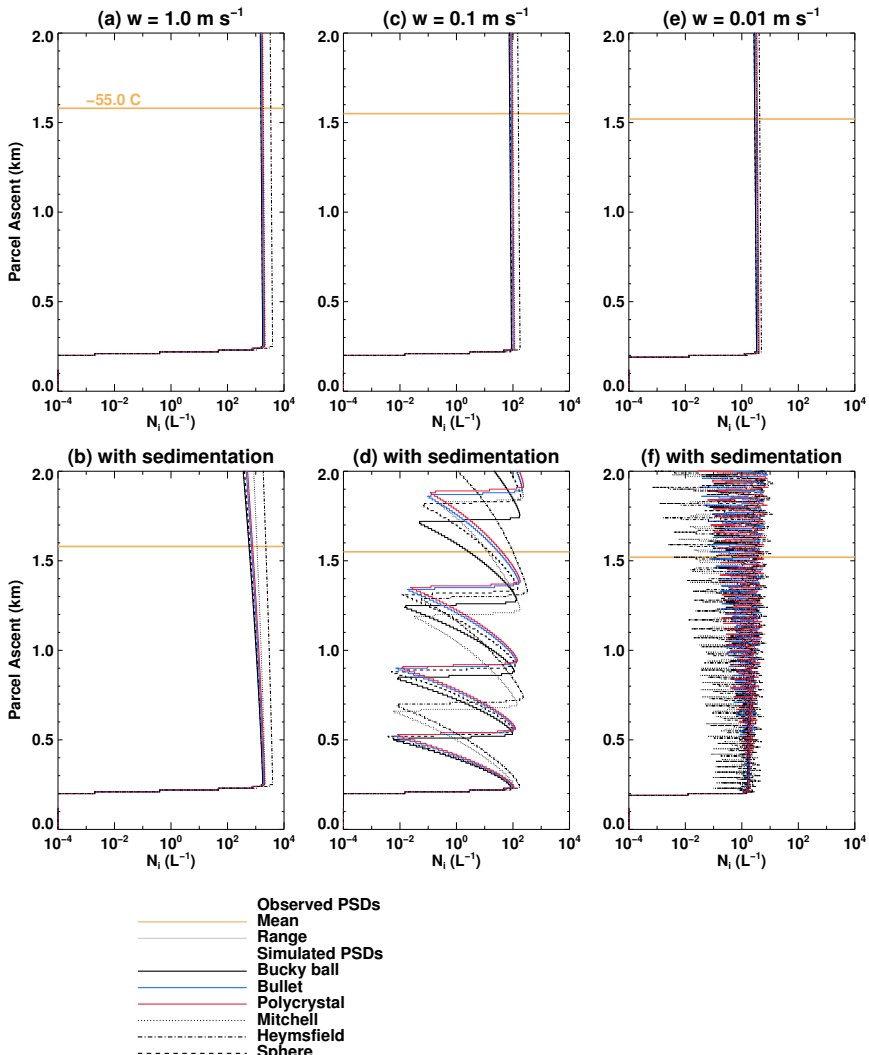

**Figure 18.** Simulated ice crystal number concentration as a function of parcel distance from initiation at $-40°$C, with ice properties as in Fig. 14 (see legend) and updraft speeds of 1, 0.1, and 0.01 m s$^{-1}$ without sedimentation (top row) and with sedimentation (bottom row). Parcel level corresponding to $-55°$C corresponding to size distributions in Fig. 17 is shown as dotted yellow line.

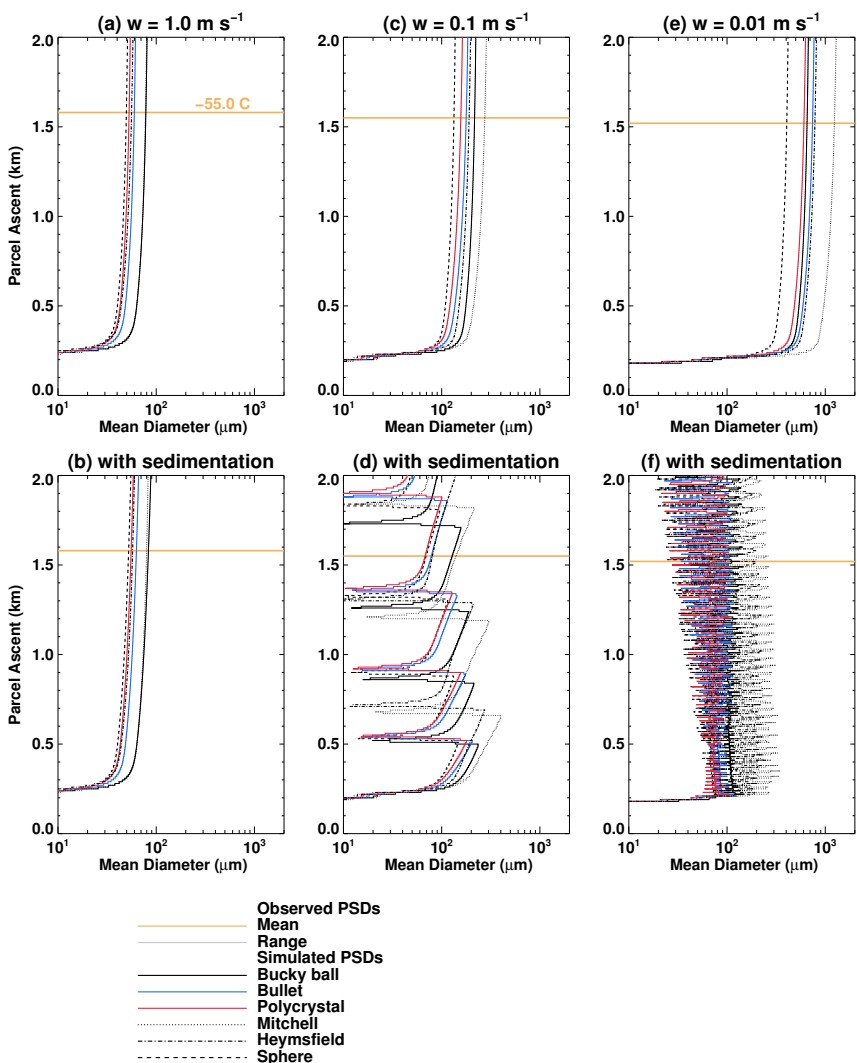

**Figure 19.** As in Fig. 18 except number-weighted mean ice crystal diameter.

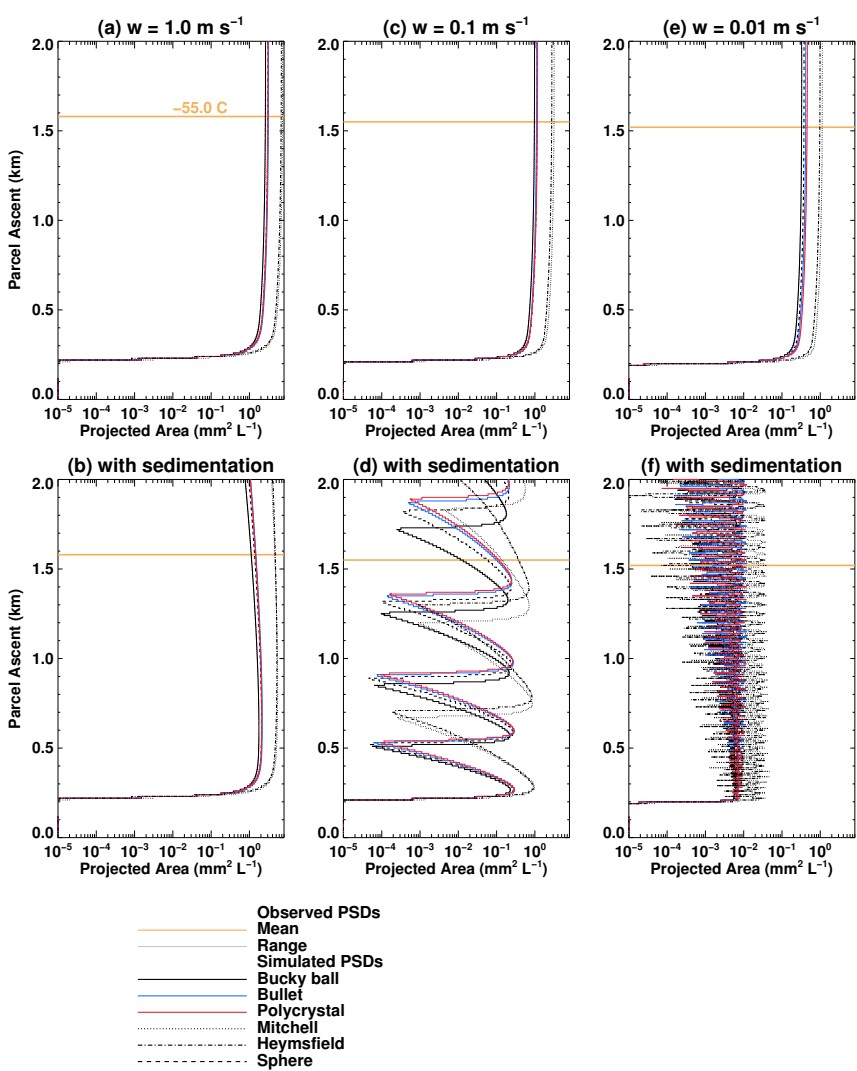

**Figure 20.** As in Fig. 18 except total ice crystal projected area.

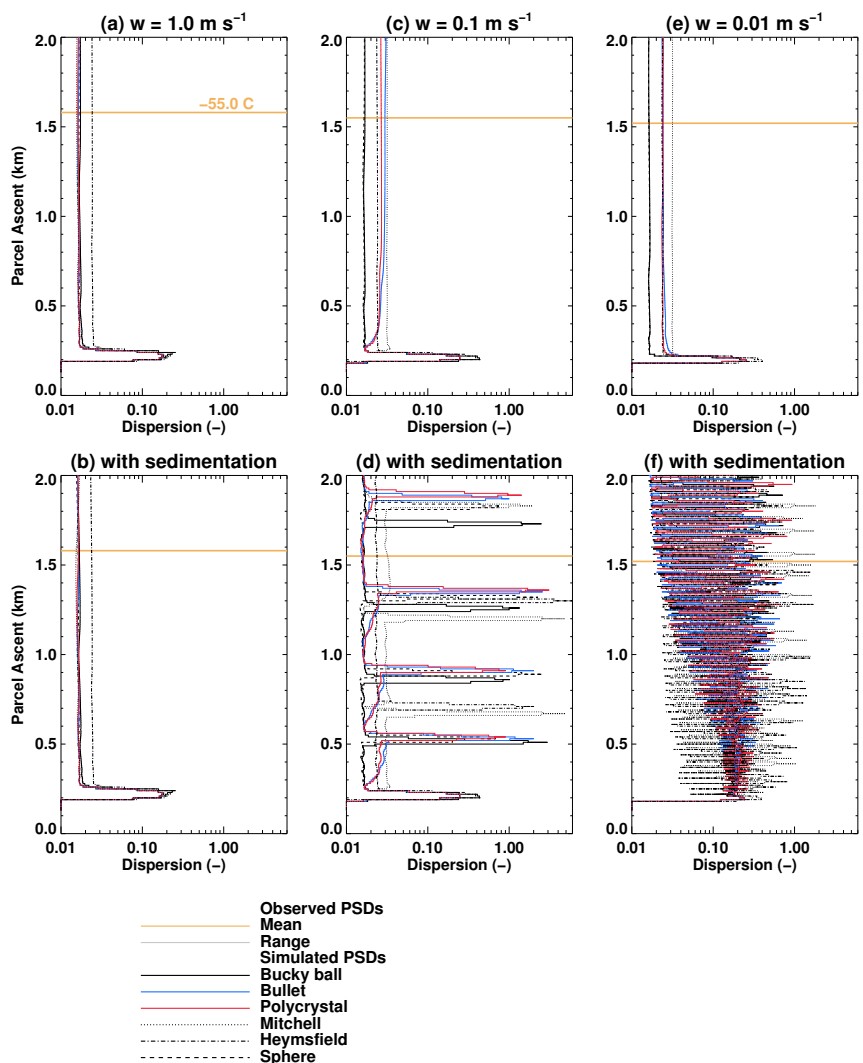

**Figure 21.** As in Fig. 18 except relative dispersion of the ice crystal size distribution.

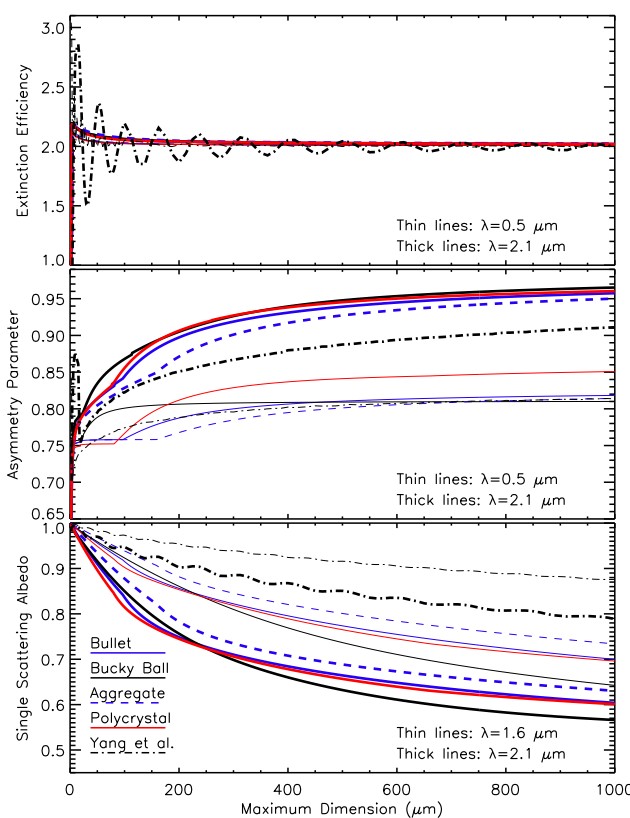

**Figure 22.** Ice single-crystal optical properties as a function of maximum dimension for the bullet, Bucky ball, aggregate, and polycrystal models, and for Yang et al. (2013) bullet rosettes (see line types in legend) at scattering and absorbing wavelengths (thin and thick lines in panels a and b) or two absorbing wavelengths (thin and thick lines in panel c).

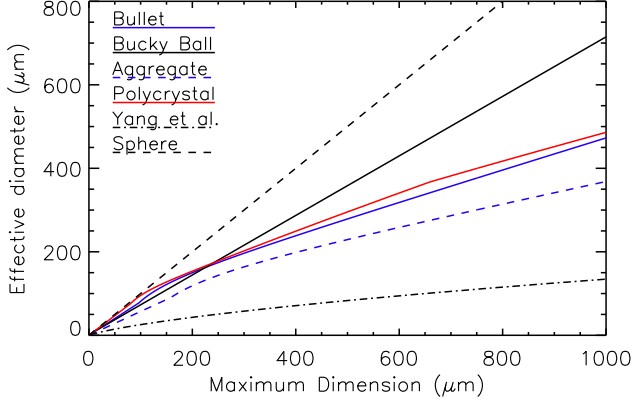

**Figure 23.** Ice single-crystal effective diameter as a function of maximum dimension for the bullet, Bucky ball, aggregate, and polycrystal models, Yang et al. (2013) bullet rosettes, and spheres (see legend).

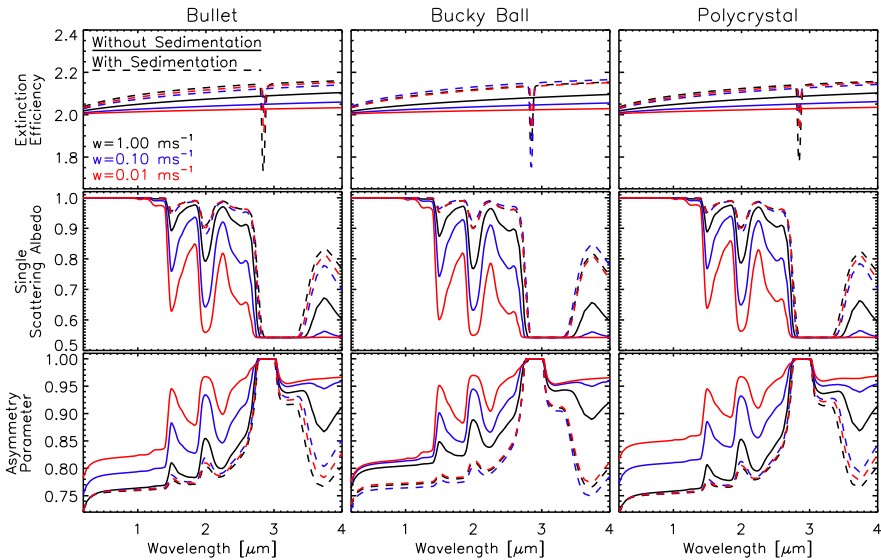

**Figure 24.** Optical properties of ice crystal size distributions shown in Fig. 17, as simulated at $-55°$C with varying ice properties (bullet, Bucky ball and polycrystal: left to right) and varying updraft speeds (line colors per legend), with and without sedimation (solid and dashed lines per legend).