# Peer review of "Derivation of physical and optical properties of midlatitude cirrus ice crystals for a size-resolved cloud microphysics model"

_Atmospheric Chemistry and Physics, 2015_

## Referee Comment (RC1) · Anonymous Referee #1 · 20 Feb 2016

**General Comments:**

This study presents a novel approach in estimating ice particle properties from CPI data, and is worthy of publication in ACP after suitable revisions. It is well organized and written, with figures of good quality.

A good effort is made to compare the new ice properties with selected properties published 20 or 14 years ago, but no analytical expressions are given for the new ice properties. A table should be added to the paper, similar to Table 1, but showing the mass- and area-dimensional coefficients for the new m-D and A-D relationships (based on CPI data); like results for the new bullet rosettes, bucky-balls, and the polycrystal model. That will allow the community to compare these new results in future studies of

ice properties, and promote progress in this field.

It would also be a service to the community if the recent work of Erfani and Mitchell (ACPD, 2015) were compared against the m-D and A-D results from this study. The Erfani-Mitchell expressions are not for a specific ice crystal habit, but were derived from a mixture of habits (similar to the Cotton et al. m-D results featured in this paper).

In Lawson et al. (2006, JAS), Sec. 3d, A-D power laws are given for irregulars, bullet rosettes, budding rosettes, and rimed rosettes with slide plane wafers between the branches (similar to the polycrystal model developed in this study). It would be instructive to compare these new results against those A-D expressions since they were also based on CPI data.

This study apparently applies a single ice crystal model over the entire ice particle size distribution (PSD). However, this assumption is questionable based on CPI observations; see Lawson et al., 2006, JAS, Fig. 5. There you can see that, with increasing size and mass-weighted percent, the smallest crystals tend to be quasi-spherical, then small irregulars, then small irregulars and budding rosettes, then larger rosettes for a single cirrus flight. While this trend may change somewhat from flight-to-flight, it is illustrative of what is typically encountered for cirrus cloud measurements. Similar results are shown in Fig. 13 of Baker and Lawson (2006, JAS). The paper should include some discussion of this, and how such size-dependent habit variation may impact the model results.

Lastly, due to the large number of symbols used in this paper, an appendix for symbol definition is recommended.

Major Comments:

1. Page 8, lines 20-21: Why is arm width W twice the hexagon side length?

2. Page 9, lines 5-6: Is there vapor competition between homo- and heterogeneous ice nucleation?
3. Page 9, lines 21-22: Are all ice nuclei composed of (NH4) HSO4?

4. Page 13, lines 14-15: How much difference is there between your Dmax and the Dmax that Mitchell uses? For random orientation, it seems that on average the branches would be oriented at ~45 degrees relative to their maximum extension. Taking that maximum length as L = 1.0 (arbitrary units) and true Dmax = 2 L, then the percent error made by Mitchell by underestimating Dmax as ~ 2L cos(45) (randomly oriented) would be ~ 29%. This seems like too small an error to account for most of the 4-fold difference in mass.

5. Page 17, line 3: Please add temperature information to Fig. 10 so that this sentence makes sense.

6. Page 19, lines 22-24: No need to wait for future studies; this information already exists (as noted under General Comments) in Lawson et al. (2006, JAS) and Baker and Lawson (2006, JAS).

7. Fig. 4. Why not use log-log plots when plotting m-D and A-D since this should be quasi-linear and make the results easier to interpret?

8. Fig. 15. Is there a super-position of the Mitchell and Heymsfield curves?

Minor Comments:

- 1. Page 11, line 11: What are "cap vertices"? Please define.
- 2. Page 20, line 18: Should < 100 be > 200?
- 3. Page 24, line 11: Does i need defining?

4. Page 26, line 13: greater => less? This is a Christiansen band where nr < 1.0 but ni is not > 1.0.

---

## Referee Comment (RC2) · Anonymous Referee #2 · 28 Mar 2016

The research effort reported in the manuscript analyzed ice crystal images in midlatitude cirrus clouds, towards developing internally consistent ice physical and optical properties for a size-resolved cloud microphysics model. Often reported in the literature, the parameterizations of ice cloud radiative properties and the counterparts of ice cloud microphysical properties are separately developed and thus lack internal consistency. The outcomes of this study represent an important contribution to a better understanding of ice cloud microphysical and radiative properties. Overall, the manuscript is well organized and clearly written. However, some improvements seem necessary before the manuscript is formally accepted for publication. Listed below are the reviewer's specific comments, which are mainly focused on the optical properties of ice crystals.

[Figure]

Several ice crystal habit models (specifically, a bucky ball model, an aggregate model, and a polycrystal model) are investigated in detail. For feasible light scattering calculation, ice crystal morphologies are highly simplified in comparison with realistic counterparts. A common justification for the simplifications is that the optical properties are realistic although ice crystal geometries are simplified and even unrealistic. An important constraint to check whether an ice crystal habit model is reasonable from the optical property perspective is to check the consistency of the corresponding optical properties between solar and infrared bands. The optical property parameterization in this study is largely based on Dr. van Diedenhoven's previous parameterizations. If the reviewer recollects correctly, Diedenhoven's previous parameterizations are developed for the solar bands, for example, van Diedenhoven et al. (2014a). Thus, it is suggested that the consistency of the present models between solar and infrared bands be validated. For the authors' information, a recent study in this regard has been reported: Holz, R.E., S. Platnick, K. Meyer, M. Vaughan, G. Wind, S. Dutcher, S. Ackerman, A. Heidinger, N. Amarasinghe, C. Wang, and P. Yang, "Resolving cirrus optical depth biases between CALIOP and MODIS using IR retrievals," Atmos. Chem. Phys. Discuss., 15, 29455-29495, doi:10.5194/acpd-15-29455-2015, 2015.

The description of the optical property simulations requires clarification. For example, it is mentioned in the manuscript (the second paragraph on page 24) that the anomalous diffraction theory (ADT) was used to compute the extinction efficiency. However, ADT is not applicable to the phase function (thus, the asymmetry factor) computation. How is an asymmetry factor value that is consistent with the ADT simulation derived?

On page 25 it is stated "a roughness parameter sigma as defined as Mack et al. (1996)..." (line 3) and "...we note that assuming plates with sigma=0.5...". In addition, Yang et al. (2013) and Baum et al. (2014) are cited. In Mack et al. (1996), uniformly tilting of ice crystal facets is assumed whereas the Gaussian distribution is assumed in Yang et al. (2013) and Baum et al. (2014). It is explicitly mentioned "Since Baum et al. (2014) and van Diedenhoven et al. (2014b) show that a roughness parameter of 0.5

best fit observations...". The same roughness parameter value (0.5) cannot be applied to the aforesaid two roughness definitions. Thus, it is suggested that an explicit definition of the roughness parameter be explicitly defined (maybe, an equation should be provided here). The clarification is important because the degree of surface roughness is a critical factor in determining the radiative forcing of ice clouds as illustrated by the following paper: Yi, B., P. Yang, B. A. Baum, T. L'Ecuyer, L. Oreopoulos, E. J. Mlawer, A. J. Heymsfield, K.-N. Liou, 2013: Influence of ice particle surface roughening on the global cloud radiative effect, J. Atmos. Sci., 70, 2794-2807.

One page 4, acronyms SHEBA and ISDAC should be spelled out.

To resolve small sizes, it is suggested that logarithmic scale is applied to the maximum dimension in Figs. 15 and 22.

---

## Author Comment (AC1) · 10 May 2016

**Response to Anonymous Referee #1**

General Comments:
This study presents a novel approach in estimating ice particle properties from CPI data, and is worthy of publication in ACP after suitable revisions. It is well organized and written, with figures of good quality.

We appreciate the positive assessment and respond to comments below.

A good effort is made to compare the new ice properties with selected properties published 20 or 14 years ago, but no analytical expressions are given for the new ice properties. A table should be added to the paper, similar to Table 1, but showing the mass- and area-dimensional coefficients for the new m-D and A-D relationships (based on CPI data); like results for the new bullet rosettes, bucky-balls, and the polycrystal model. That will allow the community to compare these new results in future studies of ice properties, and promote progress in this field.

Ice properties are supplied as supplementary material (as was already noted in the appendix). Clarification now added more prominently to close of introduction: "Because the derivations here are based on crystal component geometries and do not yield continuous analytic relationships, equations are provided in Appendix A and derived ice properties are provided for download as the Supplement."

It would also be a service to the community if the recent work of Erfani and Mitchell (ACPD, 2015) were compared against the m-D and A-D results from this study. The Erfani-Mitchell expressions are not for a specific ice crystal habit, but were derived from a mixture of habits (similar to the Cotton et al. m-D results featured in this paper).

We have added Erfani and Mitchell (now ACP, 2016) fits to Figs. 5, 7, 13 and 14, and associated discussion to Sections 1, 4.1, 4.4, 5.1 and 6.

In Lawson et al. (2006, JAS), Sec. 3d, A-D power laws are given for irregulars, bullet rosettes, budding rosettes, and rimed rosettes with slide plane wafers between the branches (similar to the polycrystal model developed in this study). It would be instructive to compare these new results against those A-D expressions since they were also based on CPI data.

We have added the Lawson et al. (2006) m-A relations for budding rosettes and rosettes to Figs. 5 and 7 and associated discussion to Sections 4.1 and 5.1. We have not addressed mixed-phase clouds or riming whatsoever in this study, so we omit rimed rosettes. Because their data set obviously represents a much wider range of particle types, including mixed-phase conditions, we also omit irregulars to guarantee applicability here.

This study apparently applies a single ice crystal model over the entire ice particle size

distribution (PSD). However, this assumption is questionable based on CPI observations; see Lawson et al., 2006, JAS, Fig. 5. There you can see that, with increasing size and mass-weighted percent, the smallest crystals tend to be quasi-spherical, then small irregulars, then small irregulars and budding rosettes, then larger rosettes for a single cirrus flight. While this trend may change somewhat from flight-to-flight, it is illustrative of what is typically encountered for cirrus cloud measurements. Similar results are shown in Fig. 13 of Baker and Lawson (2006, JAS). The paper should include some discussion of this, and how such size-dependent habit variation may impact the model results.

We have expanded the discussion intended to address uncertainty in small particle shape in the conclusions (now an independent fifth paragraph of section 6), giving greater emphasis to the concept of habit evolution from quasi-spherical shapes.

Lastly, due to the large number of symbols used in this paper, an appendix for symbol definition is recommended.

We omitted this because various symbols used only in the appendix would require long definitions, and listing the appendix symbols together with those commonly used in the main text would be unnecessarily long for most readers.

Major Comments:
1. Page 8, lines 20-21: Why is arm width W twice the hexagon side length?

This choice is convenient for equations shown in the appendix.

2. Page 9, lines 5-6: Is there vapor competition between homo- and heterogeneous ice nucleation?

Clarification added to model description of homogeneous aerosol freezing: "Heterogeneous freezing is neglected." And to conclusions: "Unlike the simplified parcel simulations shown here, 3D simulations will consider competition between homogeneous and heterogeneous freezing mechanisms and results can be robustly compared with observed ice size distributions."

3. Page 9, lines 21-22: Are all ice nuclei composed of $(NH_4) HSO_4$?

All ice crystals are formed from homogeneous aerosol freezing (see response 2).

4. Page 13, lines 14-15: How much difference is there between your Dmax and the Dmax that Mitchell uses? For random orientation, it seems that on average the branches would be oriented at 45 degrees relative to their maximum extension. Taking that maximum length as L = 1.0 (arbitrary units) and true Dmax = 2 L, then the percent error made by Mitchell by underestimating Dmax as 2L cos(45) (randomly

oriented) would be 29%. This seems like too small an error to account for most of the 4-fold difference in mass.

Clarification added in a follow-on sentence: "However, we are unable to quantitatively confirm that [differences in $m$ are primarily attributable to differing approaches to defining $D_{max}$] because randomly oriented maximum dimension cannot be calculated analytically for the idealized geometries derived here nor obtained from CPI images for the natural crystals."

5. Page 17, line 3: Please add temperature information to Fig. 10 so that this sentence makes sense.

Since Fig. 10 is not intended to be statistically representative, we have changed "some rosettes falling from colder temperatures reach a plate growth regime" to "some rosettes exhibit a plate growth regime".

6. Page 19, lines 22-24: No need to wait for future studies; this information already exists (as noted under General Comments) in Lawson et al. (2006, JAS) and Baker and Lawson (2006, JAS).

We added Lawson et al. (2006) relations for budding rosettes and rosettes to Figs. 5 and 7, as discussed above, but omitted Baker and Lawson (2006) owing to their stated conditions, i.e., "The crystal types typical of high-altitude cirrus, that is, bullet rosettes and similar spatial crystals, are not represented in this dataset."

7. Fig. 4. Why not use log-log plots when plotting m-D and A-D since this should be quasi-linear and make the results easier to interpret?

Because of our focus on measurements made over less than one order of magnitude in maximum dimension and our concern with geometric differences, we prefer linear axes when we can use them, as in most panels of Fig. 4. We prefer log axes when emphasizing the small particle size range, where we lack Ice Crystal Ruler measurements, as in Fig. 5.

8. Fig. 15. Is there a super-position of the Mitchell and Heymsfield curves?

Clarification added to caption: "In the absence of specified $\alpha_e$ for some or all crystal sizes, a constant value is taken for Mitchell et al. (1996) and Heymsfield et al. (2002) ice properties (see text)."

Minor Comments:
1. Page 11, line 11: What are "cap vertices"? Please define.

Clarification added: "opposing cap vertices" replaced with "opposing edges of the hexagonal pyramids that cap each branch", and other occurrence of "vertices" replaced with "edges" in the second sentence of Appendix A1.

2. Page 20, line 18: Should < 100 be > 200?

Here < should have been > (correction made), thank you.

3. Page 24, line 11: Does i need defining?

The i was defined in Section 4.1 (bulk density of ice).

4. Page 26, line 13: greater => less? This is a Christiansen band where nr < 1.0 but ni is not > 1.0.

Indeed "greater" should have been "less" and we thank the reviewer for pointing out that this is a Christiansen band. We changed the text to read "At ~2.8 micron, a Christiansen band (Arnott et al. 1995) is present where a combination of strong absorption and refractive indices near or less than unity leads to a decrease in $Q_e$ (cf. Baum et al. 2014)."

---

## Author Comment (AC2) · 10 May 2016

**Response to Anonymous Referee #2**

The research effort reported in the manuscript analyzed ice crystal images in mid-latitude cirrus clouds, towards developing internally consistent ice physical and optical properties for a size-resolved cloud microphysics model. Often reported in the literature, the parameterizations of ice cloud radiative properties and the counterparts of ice cloud microphysical properties are separately developed and thus lack internal consistency. The outcomes of this study represent an important contribution to a better understanding of ice cloud microphysical and radiative properties. Overall, the manuscript is well organized and clearly written. However, some improvements seem necessary before the manuscript is formally accepted for publication. Listed below are the reviewer's specific comments, which are mainly focused on the optical properties of ice crystals.

We appreciate the positive assessment and respond to comments below.

Several ice crystal habit models (specifically, a bucky ball model, an aggregate model, and a polycrystal model) are investigated in detail. For feasible light scattering calculation, ice crystal morphologies are highly simplified in comparison with realistic counterparts. A common justification for the simplifications is that the optical properties are realistic although ice crystal geometries are simplified and even unrealistic. An important constraint to check whether an ice crystal habit model is reasonable from the optical property perspective is to check the consistency of the corresponding optical properties between solar and infrared bands. The optical property parameterization in this study is largely based on Dr. van Diedenhoven's previous parameterizations. If the reviewer recollects correctly, Diedenhoven's previous parameterizations are developed for the solar bands, for example, van Diedenhoven et al. (2014a). Thus, it is suggested that the consistency of the present models between solar and infrared bands be validated. For the authors' information, a recent study in this regard has been reported: Holz, R.E., S. Platnick, K. Meyer, M. Vaughan, G. Wind, S. Dutcher, S. Ackerman, A. Heidinger, N. Amarasinghe, C. Wang, and P. Yang, "Resolving cirrus optical depth biases between CALIOP and MODIS using IR retrievals," Atmos. Chem. Phys. Discuss., 15, 29455-29495, doi:10.5194/acpd-15-29455-2015, 2015.

As infrared radiative transfer is dominated by emission, crystal shape has relatively little influence on it. In Holz et al. (2016 now in ACP) this fact is used to evaluate the applicability of assumed habit for shortwave retrievals. Holz et al. state that "because the sensitivity of IR IOT retrievals to ice crystal habit selection is minimal, these retrievals provide an independent means to evaluate the CALIOP and MODIS solar reflectance retrievals." Thus, an evaluation as presented by Holz et al. compares retrievals of optical thickness (and/or particle size) from both shortwave and IR measurements in order to evaluate of the optical model used for the shortwave. In contrast, our investigation aims to derive optical properties consistent with the in situ observations at various levels within cloud. Representative shortwave and IR measurements for the particular clouds under investigation do not exist and an evaluation as performed by Holz et al. therefore cannot be performed for these clouds. To clarify, we added

the following text to line 28 on page 23: "Infrared radiative transfer is dominated by emission, which is affected by particle size, but its sensitivity to crystal shape is minimal (e.g., Holz et al. 2016). However, particle shape does affect the relevant shortwave optical properties substantially."

The description of the optical property simulations requires clarification. For example, it is mentioned in the manuscript (the second paragraph on page 24) that the anomalous diffraction theory (ADT) was used to compute the extinction efficiency. However, ADT is not applicable to the phase function (thus, the asymmetry factor) computation. How is an asymmetry factor value that is consistent with the ADT simulation derived?

Since van Diedenhoven et al. (2014) is based on geometric optics calculations, it assumes extinction efficiency of 2 for all particles and wavelengths, and the anomalous diffraction was only used to partly correct this simplification for small particle sizes. Clarification added at page 24, line 13: "The van Diedenhoven et al. (2014) parameterization is based on geometric optics calculations. Accordingly, it assumes the extinction efficiency (Qe) to be 2 for all particles and wavelengths. To partly correct this simplification for small particle sizes, here we apply anomalous diffraction …"

On page 25 it is stated "a roughness parameter sigma as defined as Mack et al. (1996) …" (line 3) and "…we note that assuming plates with sigma=0.5…". In addition, Yang et al. (2013) and Baum et al. (2014) are cited. In Mack et al. (1996), uniformly tilting of ice crystal facets is assumed whereas the Gaussian distribution is assumed in Yang et al. (2013) and Baum et al. (2014). It is explicitly mentioned "Since Baum et al. (2014) and van Diedenhoven et al. (2014b) show that a roughness parameter of 0.5 best fit observations…". The same roughness parameter value (0.5) cannot be applied to the aforesaid two roughness definitions. Thus, it is suggested that an explicit definition of the roughness parameter be explicitly defined (maybe, an equation should be provided here). The clarification is important because the degree of surface roughness is a critical factor in determining the radiative forcing of ice clouds as illustrated by the following paper: Yi, B., P. Yang, B. A. Baum, T. L'Ecuyer, L. Oreopoulos, E. J. Mlawer, A. J. Heymsfield, K.-N. Liou, 2013: Influence of ice particle surface roughening on the global cloud radiative effect, J. Atmos. Sci., 70, 2794-2807.

Various definitions of the roughness parameters were compared by Neshyba et al. (2013) and by Geogdzhayev and van Diedenhoven (2016) and were found to be largely equivalent. This means that the same value of roughness parameter defined as by Macke et al. and that used by Baum et al. (2014) yields largely equivalent scattering properties. As demonstrated by Geogdzhayev and van Diedenhoven (2016), a roughness parameter of a given value but with different definitions represent very similar micro-structures on the crystal surfaces. As stated in the submitted manuscript (page 25, line 1): "In the van Diedenhoven et al. (2014a) parameterization, the level of surface distortion is specified by a roughness parameter δ as defined by Macke et al. (1996); differently defined roughness parameters are found to be

roughly equivalent (Neshyba et al. , 2013; Geogdzhayev and van Diedenhoven, 2016)." This text is now extended to read:

"In the van Diedenhoven et al. (2014a) parameterization, the level of surface distortion is specified by a roughness parameter δ as defined by Macke et al. (1996). The Macke et al. (1996) ray-tracing code perturbs the normal of the crystal surface from its nominal orientation by an angle that, for each interaction with a ray, is varied randomly with uniform distribution between 0 and delta times 90°. Similar commonly used parameterizations of particle roughness perturb the crystal surfaces using Weibull (Shcherbakov et al. 2006) or Gaussian (Baum et al. 2014) statistics rather than uniform distributions. However, Neshyba et al. (2013) and Geogdzhayev and van Diedenhoven (2016) demonstrated that the same roughness parameter value defined through a Weibull, Gaussian or uniform distribution represents very similar crystal microscale surfaces and yields largely equivalent scattering properties."

One page 4, acronyms SHEBA and ISDAC should be spelled out.

Now spelled out.

To resolve small sizes, it is suggested that logarithmic scale is applied to the maximum dimension in Figs. 15 and 22.

The choice of axes throughout is debatable. Here we prefer to use the same linear axis consistently across Figs. 14, 15, 22 and 23.